# Heterogeneous multi-compartmental hydrogel particles as synthetic cells for incompatible tandem reactions

Hongliang Tan[1], Song Guo[1], Ngoc-Duy Dinh [1], Rongcong Luo[1], Lin Jin[1] & Chia-Hung Chen[1,2,3]

In nature, individual cells contain multiple isolated compartments in which cascade enzymatic reactions occur to form essential biological products with high efficiency. Here, we report a cell-inspired design of functional hydrogel particles with multiple compartments, in which different enzymes are spatially immobilized in distinct domains that enable engineered, one-pot, tandem reactions. The dense packing of different compartments in the hydrogel particle enables effective transportation of reactants to ensure that the products are generated with high efficiency. To demonstrate the advantages of micro-environmental modifications, we employ the copolymerization of acrylic acid, which leads to the formation of heterogeneous multi-compartmental hydrogel particles with different pH microenvironments. Upon the positional assembly of glucose oxidase and magnetic nanoparticles, these hydrogel particles are able to process a glucose-triggered, incompatible, multistep tandem reaction in one pot. Furthermore, based on the high cytotoxicity of hydroxyl radicals, a glucose-powered therapeutic strategy to kill cancer cells was approached.

[1] Department of Biomedical Engineering, National University of Singapore, Singapore 117583, Singapore. [2] Singapore Institute for Neurotechnology (SINAPSE), Singapore 117456, Singapore. [3] Biomedical Institute for Global Health Research and Technology, Singapore 117599, Singapore. Correspondence and requests for materials should be addressed to C.-H.C. (email: biecch@nus.edu.sg)

Living cells contain multiple compartmentalized organelles surrounded by membranes that perform distinct functions to maintain cell physiology. The close confinement and dense packing of different organelles can facilitate cascade enzymatic bioreactions with efficient bioproduct transportation between compartments, minimizing the decomposition of active intermediates[1]. Moreover, the presence of organelle compartments enables the spatial confinement of incompatible and opposing reagents in different cellular domains, shielding them from each other and allowing their corresponding reactions to occur in optimized microenvironments[2, 3]. Accordingly, numerous incompatible (and competing) catalytic transformations can be performed simultaneously in a cell with unsurpassed efficiency and specificity, producing system-level essential bioproducts.

Recently, the construction of multi-compartmental systems to perform distinct biochemical reactions in one pot, as in living cellular systems, has attracted substantial attention[4–7]. Soft components, such as liposomes, polymersomes, and polymer capsules, have been investigated for the fabrication of various types of multi-compartmentalized systems, including vesosomes[8], polymersome-in-polymersome structures[9], and capsosomes[10]. However, the contradiction between close confinements with dense packing and micro-environmental controls of different compartments remained. For example, the liposomes and polymersomes were developed to closely immobilize enzymes for dense packing confinement, triggering cascade process effectively, but these systems reached the limitations in processing individual reactions in spatial compartmental microenvironments[11–13]. The Pickering emulsions and functional polymeric micelles were manipulated to form distinguished spatial compartments to optimize the incompatible tandem reactions, but without well-defined close packing, the product generation efficiency was limited by slow transportations of intermediates between the compartments in Pickering emulsions and polymeric micelles[14, 15]. Moreover, these systems required organic chemical processes to prepare, causing the challenges in bio-compatibility.

Hydrogels are three-dimensionally cross-linked polymeric networks containing ~ 95% water, similar to natural cells. On the basis of the high porosity of hydrogels, functional enzymes can be immobilized in the network to produce enzymatic products over long periods[16, 17]. Compared with free enzymes, encapsulated enzymes often show greatly improved catalytic activities, enhanced thermal stabilities and tolerance for both high-pH conditions and organic solvents[18, 19]. Moreover, hydrogels exhibit reliable mechanical stability and biocompatibility, making them widely applicable in the biomedical industry. In this regard, hydrogels based systems appear promising for the construction of multi-compartmental systems. Several methods have been developed to fabricate structured hydrogels with multiple compartments[20–22] to load different drugs in separate compartments with a programmed, controllable sequential release. However, most such studies focused on controlling drug release from hydrogels, requiring a drug-reloading process for long-term drug delivery. These hydrogels were not used for enzyme-triggered product fabrication, as occurs in cells. Indeed, for the incorporation of multistep enzymatic reactions and the system-level production, it is essential to construct well-defined multi-compartmental cell-like hydrogel systems with immobilized enzymes.

In this study, a heterogeneous multi-compartmental hydrogel particle with distinct microenvironments was designed and fabricated using microfluidics within which incompatible multistep tandem reactions were conducted. The combination of close confinement of hydrogel building blocks and distinguished compartments through microfluidic assembly acted synergistically to perform tandem reactions. The dense package of hydrogels in micro-scaled allowed effective transportation of reactants, while the distinguished compartments offered desirable micro-environment controls to ensure product generation with high efficiency. For example, the copolymerization of acrylic acid (AA) provided an acidic microenvironment for the corresponding hydrogel particles because of the presence of a carboxyl acid group. The incompatible natural enzyme and nanozyme were loaded independently in defined compartments during the formation of these hydrogel particles through microfluidics, allowing the spatial organization and segregation of the incompatible multistep tandem reaction. Accordingly, the incompatible multistep tandem reaction occurred simultaneously within the multi-compartmentalized hydrogel particles without mutual interference. The products generated by the enzymatic reactions in the first compartment were effectively transported to the second compartment, which contained a distinct environment, triggering catalysis reactions to accomplish a system-level tandem reaction with high efficiency in one pot.

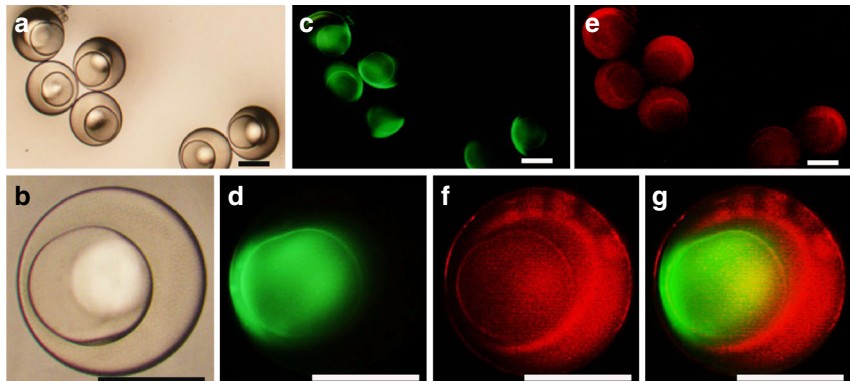

**Fig. 1** Preparation of multi-compartmental hydrogel particles using microfluidics. **a** The bright-field image of polyethylene glycol (PEG) multi-compartmental hydrogel particles was recorded. **b** The bright-field image of a single particle was recorded to observe its details. **c** The cores of the PEG multi-compartmental hydrogel particles were labeled with Albumin-FITC, showing *green florescence* signal with emission wavelength 520 nm. **d** The image of a single particle, whose core was labeled by Albumin-FITC was recorded. **e** The shells of the PEG multi-compartmental hydrogel particles were labeled with Dextran-RhB, showing *red florescence* signal with emission wavelength 590 nm. **f** The image of a single particle, whose shell was labeled by Dextran-RhB was recorded. **g** The image of a particle with core labeled with Albumin-FITC (*green fluorescence*) and shell labeled with Dextran-RhB (*red fluorescence*) was recorded. *Scaling bars* are 200 μm

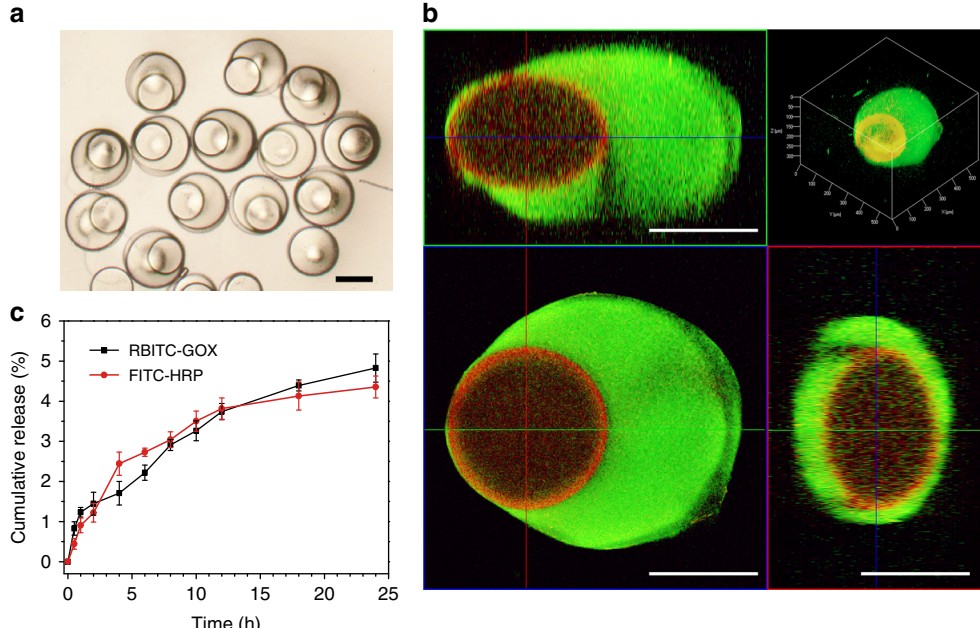

**Fig. 2** Encapsulation of GOX and HRP in a multi-compartmental particle. **a** Bright-field microscopic image of GOX@HRP. **b** Confocal scanning laser microscopic images of GOX@HRP in x–y, x–z and y–z planes, along with a 3D model of GOX@HRP. The GOX and HRP were labeled with rhodamine B isothiocyanate (RBITC, *red*) and fluorescein isothiocyanate (FITC, *green*), respectively. Detailed labeling procedures are discussed in Supplementary Methods (Labeling of enzymes with fluorescent dyes). **c** GOX and HRP release profiles from GOX@HRP in PBS (10 mM, pH 7.0). *Scaling bars* are 200 μm

## Results

**Fabrication of multi-compartmental hydrogel particles.** A two-step sequential gelation process along with droplet assembly in a microfluidic device was conducted to fabricate multi-compartmental hydrogel particles, as shown in Supplementary Fig. 1. The prepared PEG-diacrylate (PEGDA, Mn = 575) hydrogel particles (diameter ~ 300 μm) were monodisperse with well-defined cores (diameter ~ 200 μm). The cores and shells of these particles were loaded with albumin labeled with fluorescein isothiocyanate (Albumin-FITC) and dextran labeled with rhodamine B isothiocyanate (Dextran-RhB), respectively. Bright-field image of these particles is shown in Fig. 1a. The magnified bright-field image (Fig. 1b) clearly exhibits the core–shell structure in the multi-compartmental hydrogel particle. With Albumin-FITC loaded in the cores of these particles, under excitation light (wavelength: 488 nm), a green florescence signal (wavelength: 520 nm) was emitted from the cores, shown in Fig. 1c. The magnified image of the particle core is shown in Fig. 1d. With Dextran-RhB loaded in the shells of these particles, under excitation light (wavelength: 560 nm), a red florescence signal (wavelength: 590 nm) was emitted from the shells, as shown in Fig. 1e. The magnified image of the particle shell is shown in Fig. 1f. An image combining the fluorescence signals from the both core and shell of the particle is shown in Fig. 1g. Hence, the microfluidic strategy was suitable for fabricating multi-compartmental hydrogel particles, in which different contents were spatially confined in distinct domains.

**Enzyme encapsulation and characterization.** Glucose oxidase (GOX) and horseradish peroxidase (HRP) were selected as model enzymes for performing tandem reaction, which has already been applied as an extremely popular system in various compartmentalized structures[4–6, 12, 13]. The single-compartmental PEG hydrogel particles were prepared as the carriers for enzyme encapsulation. Two different single-compartmental particles containing GOX (GOX@PEG) and HRP (HRP@PEG) were fabricated (Supplementary Fig. 2). The encapsulation efficiencies of GOX and HRP in PEG hydrogel were determined as ~ 75% and 83%, respectively, using bicinchoninic acid (BCA) assays. The catalytic activities of the enzymes were not affected by the encapsulation of the enzymes in PEG hydrogel particles. It was found that the catalytic activities of GOX@PEG and HRP@PEG were slightly higher than the activities of free GOX (~ 5.5%) and HRP (~ 2.1%) when adding the same amount of free enzymes in the same reaction volume (Supplementary Fig. 3a). This enhancement was caused by the high local concentration of enzymes in the PEG hydrogel particles, which effectively triggered catalytic reactions. The enzyme-releasing rate from PEG hydrogel particles was quantitatively characterized, revealing that ~ 8.3% of GOX and 9.9% of HRP were released from the hydrogel particles within 24 h (Supplementary Fig. 3b). On the basis of these facts, PEG hydrogel is an ideal component to construct multi-compartmental structures for the incorporation of tandem enzymatic reactions.

The multi-compartmental hydrogel particles encapsulating different enzymes in distinguished compartments were prepared to process multistep tandem reactions. GOX and HRP were loaded into the inner and outer compartments of the hydrogel particles, respectively, to create a multi-compartmental system denoted as GOX@HRP (Fig. 2a). The multi-compartmental particles with two distinct domains loaded by RBITC-GOX (red color, core) and FITC-HRP (green color, shell) were observed using a confocal scanning laser microscope (Fig. 2b). In addition to fluorescent imaging, the presence of GOX and HRP was further verified by their corresponding specific assays: the gluconic acid-$Fe^{3+}$-hydroxamine colorimetric assay for GOX (Supplementary Figs. 4 and 5)[23] and the $H_2O_2$-Amplex Red colorimetric assay for HRP (Supplementary Figs. 6 and 7)[24].

By separately measuring the total protein (Supplementary Fig. 8) and HRP (Supplementary Fig. 9) contents in GOX@HRP, the effective GOX:HRP ratio in GOX@HRP was calculated as 0.9:1. The similar amounts of GOX and HRP suggested that the inner and outer compartments of the multi-compartmental particle showed similar capabilities for enzyme immobilization.

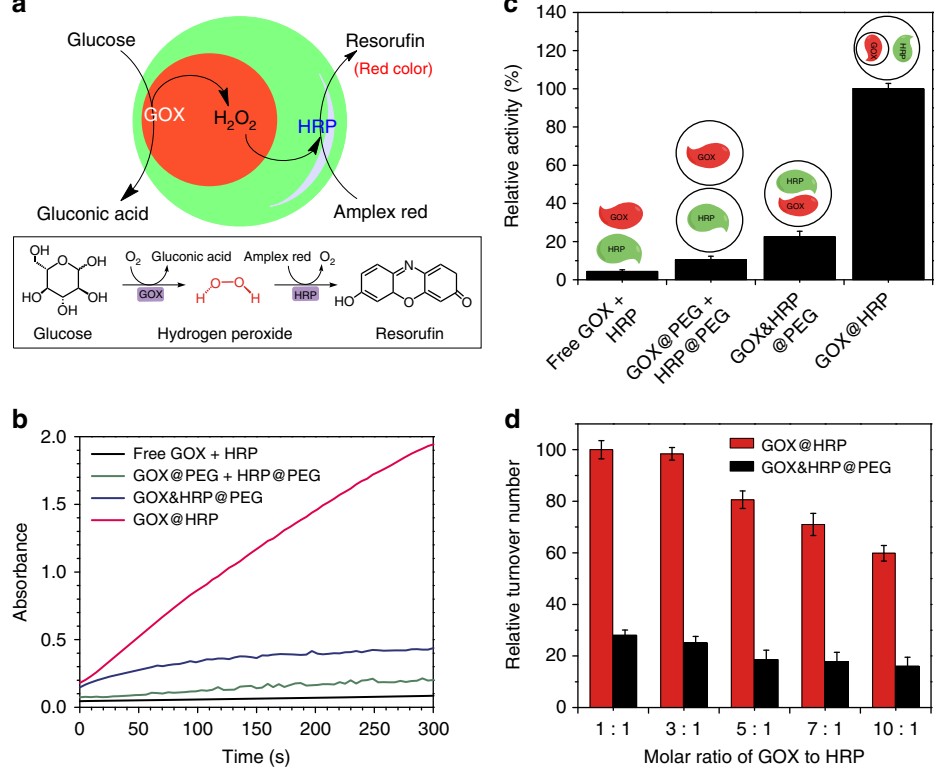

**Fig. 3** Activity characterization of GOX@HRP. **a** Schematic illustration and reaction equations of GOX@HRP-catalyzed Amplex Red oxidation for the production of resorufin after initiation by glucose. **b** Time-dependent absorbance changes caused by the oxidation product of Amplex Red catalyzed by different tandem catalytic systems: a free GOX/HRP system, a GOX@PEG/HRP@PEG system, GOX&HRP@PEG, and GOX@HRP. **c** Relative activities of tandem catalytic systems based on a free GOX/HRP system, a GOX@PEG/HRP@PEG system, GOX&HRP@PEG, and GOX@HRP. **d** Relative turnover numbers of GOX@HRP and GOX&HRP@PEG system with molar ratios of GOX to HRP from 1:1 to 10:1

Moreover, the low-release rates of GOX ( ~ 4.8%) and HRP ( ~ 4.3%) from GOX@HRP suggested that GOX and HRP were well separated in the distinguished compartments of a hydrogel particle with limited cross contamination (Fig. 2c). In addition, the kinetic behaviors of GOX and HRP in GOX@HRP were investigated separately and were compared with free GOX and HRP. Supplementary Table 1 shows that the separate GOX and HRP in GOX@HRP displayed a comparable Michaelis–Menten constant ($K$m) and turnover number ($K$cat) to those in free GOX and HRP enzymes. This finding indicates that the intermediates ($H_2O_2$) from the GOX reaction freely diffused across the multi-compartmentalized hydrogel particle without requiring artificial transport mediators[12, 13], endowing GOX@HRP with the ability to process a tandem reaction in one pot with high efficiency.

**Tandem reaction catalyzed by GOX@HRP system**. The catalytic efficiency of GOX@HRP was evaluated by the absorbance of resorufin generated through $H_2O_2$-HRP reactions. First, upon the addition of glucose, the GOX@HRP-catalyzed tandem reaction was triggered. This reaction converted glucose into gluconic acid and $H_2O_2$ through a GOX-mediated oxidation reaction. Next, the formed $H_2O_2$ reacted with HRP to oxidize Amplex Red, producing a colored product, resorufin (Fig. 3a). Because resorufin exhibited a maximum absorption peak with a wavelength of 560 nm, the tandem reaction could be monitored by the absorbance change of resorufin at 560 nm. Figure 3b shows that the Amplex Red produced by GOX@HRP induced-oxidization was time-dependently increased after the addition of glucose. A similar increase of Amplex Red was observed in the presence of the free GOX/HRP system, the GOX@PEG/HRP@PEG system

and GOX&HRP@PEG, indicating that these tandem catalytic systems shared the same reaction pathway. However, compared with GOX@HRP, a significant impedance in the absorbance of resorufin was observed in the free GOX/HRP system, the GOX@PEG/HRP@PEG system and GOX&HRP@PEG. In Fig. 3c, GOX@HRP showed the highest catalytic activity, which was ~ 23-, 10-, and 4-fold higher than that of the free GOX/HRP system, the GOX@PEG/HRP@PEG system and GOX&HRP@PEG, respectively.

A detailed kinetic analysis of GOX@HRP was performed using glucose as the substrate in Supplementary Fig. 10. The initial reaction rate of GOX@HRP versus the glucose concentration follows the typical Michaelis-Menten behavior. On the basis of the Lineweaver–Burk plots (inset of Supplementary Fig. 10d), the maximum initial velocity ($V$max) and $K$m values of GOX@HRP were calculated ($V$max = $12.491 \times 10^{-8}$ M s$^{-1}$, $K$m = 0.635 mM). Table 1 shows that the $K$m value of GOX@HRP was similar to the $K$m values of the free GOX/HRP system, the GOX@PEG/HRP@PEG system and GOX&HRP@PEG, indicating their comparable affinity to that of glucose. However, a significantly enhanced $V$max (ranging from 3.7- to 8.5-fold compared with the

**Table 1 Kinetic data for different tandem catalytic systems**

| Catalyst | Substrate | $K_m$/ (mM) | $V_{max}$ ($10^{-8}$ M s$^{-1}$) | Kcat (s$^{-1}$) |
|---|---|---|---|---|
| Free GOX + HRP | Glucose | 0.846 | 1.476 | 172.093 |
| GOX@PEG + HRP@PEG | Glucose | 0.798 | 1.724 | 200.447 |
| GOX&HRP@PEG | Glucose | 0.713 | 3.386 | 393.691 |
| GOX@HRP | Glucose | 0.635 | 12.491 | 1452.384 |

other three tandem reaction systems) and the associated increase of $K$cat were observed in GOX@HRP, demonstrating the highest catalytic performance of GOX@HRP. Since the intermediate ($H_2O_2$) could freely move across the hydrogel particle due to its small size ($\sim 0.4$ nm in diameter) and charge neutrality, the highest activity of GOX@HRP compared with the other three tandem systems was attributed to its unique multi-compartmental structure in which GOX and HRP were separately confined and closely packaged, facilitating $H_2O_2$ transfer among enzymes (also known as the proximity effect) and minimizing $H_2O_2$ inhibition to HRP activity.

**Proximity effect**. To investigate the proximity effect in GOX@HRP, the time-dependent fluorescence change of Amplex Red in GOX@HRP was recorded to compare with that in the free GOX/HRP system. Supplementary Fig. 11 shows that compared with GOX@HRP, the free GOX/HRP system showed an obvious 3-min lag phase before the fluorescence increased, suggesting the slower conversion of Amplex Red. Similar patterns were also observed in the colorimetric reactions using 3,3′,5,5′-tetramethylbenzidine (TMB) and 2,2′-azino-bis(3-ethylbenzothiazoline-6-sulfonic acid) (ABTS) as substrates (Supplementary Fig. 12), implying the universal enhanced catalytic performance of GOX@HRP. Because the GOX in GOX@HRP showed a similar affinity and reaction rate to those of glucose relative to that of the free GOX system, the slower conversion of chromogenic substrates catalyzed by the free GOX/HRP system was attributed to the period required for the transfer of $H_2O_2$ from GOX to solution and then to HRP. As expected, the GOX@PEG/HRP@PEG system without the proximity effect showed a lower catalytic activity than the activity in GOX@HRP (Fig. 3c), which confirmed that the close packing of GOX and HRP in a multi-compartmental particle leads to the effective transport of $H_2O_2$, enhancing the catalytic activity of GOX@HRP. The concentration gradient of $H_2O_2$ in a multi-compartmental hydrogel particle, causing positional effects was evaluated by using Brownian motion model[25, 26], as shown in Supplementary Fig. 13.

**Minimization of $H_2O_2$ inhibition**. Interestingly, compared with the catalysis activities of GOX@HRP and GOX&HRP@PEG (enzymes mixed homogeneously in single-PEG particles with the closest distance between GOX and HRP), a $\sim 4$-fold lower activity in GOX&HRP@PEG was observed (Fig. 3c). Previous reports have demonstrated that a high-local concentration of $H_2O_2$ suppressed the HRP activity, leading to irreversible inactivation[27–29]. In this regard, the lower reaction activity of GOX&HRP@PEG was ascribed to the inhibition of HRP activity by the high local concentration of $H_2O_2$ generated from glucose oxidization. Therefore, to optimize the catalysis activities of GOX/HRP, there are two requirements: first, the distance between GOX and HRP should be close enough to allow the effective transfer of $H_2O_2$, thus triggering tandem reactions. Second, a certain distance between GOX and HRP should be maintained to minimize $H_2O_2$ inhibition to HRP catalysis reactions[30].

To discuss the inhibition of $H_2O_2$ on the presented GOX-HRP tandem reactions, a series of glucose conversion experiments catalyzed by changing molar ratios of GOX to HRP from 1:1 to 10:1 were conducted in GOX@HRP systems, as local $H_2O_2$ concentrations were determined by GOX concentrations. As the GOX amounts in GOX@HRP increased, the oxidation reactions of Amplex Red were gradually suppressed (Supplementary Fig. 14a), which is associated with a decrease in the turnover numbers (Fig. 3d), suggesting the inhibition effect of a high local $H_2O_2$ concentration to HRP activity. Similarly, the inhibition

effect of $H_2O_2$ was also found in the GOX&HRP@PEG system (Fig. 3d and Supplementary Fig. 14b). However, compared with GOX@HRP, the GOX&HRP@PEG exhibited much lower turnover numbers under identical conditions, indicating that the inhibition of $H_2O_2$ (on the HRP activity) in GOX&HRP@PEG was stronger than that in GOX@HRP. Accordingly, a certain distance between GOX and HRP is necessary to ensure that HRP, with high activity, triggers the tandem reaction by minimizing the $H_2O_2$ inhibition effect. In addition, the dilution of the local $H_2O_2$ concentration at the HRP site due to the free and non-targeted diffusion of $H_2O_2$ in the hydrogel particle could also contribute to the enhancement of the GOX@HRP activity.

In summary, the combination of the advantages in positional assembly and spatial segregation of GOX and HRP in a micro-scale multi-compartmental system acted synergistically to perform tandem reactions. The $H_2O_2$ produced from the GOX reaction in the inner compartment was effectively transferred to the outer compartment to trigger the HRP reaction, while the HRP activity was not suppressed due to the direct interaction with a high concentration of $H_2O_2$, as occurs in living cells.

**Enzyme stability**. The enzymes in GOX@HRP displayed excellent stability and recyclability. The ability of the enzyme to be stored for a long period at room temperature ($25\,^\circ C$) was observed in Supplementary Fig. 15. In the free GOX/HRP system, $\sim 17\%$ of activity was retained after 7 days of storage in a water solution, while in GOX@HRP, $\sim 83\%$ of activity was retained under the same conditions. The thermal stability of the enzymes in GOX@HRP was tested at $60\,^\circ C$ for 60 min, as shown in Supplementary Fig. 16. After the thermal treatment, $> 65\%$ of the activity of GOX@HRP was retained, whereas $> 96\%$ of the activity of the free GOX/HRP pair was lost. This result indicated that the thermal stability of GOX@HRP was enhanced, possibly because of the decelerated enzyme denaturation caused by aggregation[31, 32]. To test the biological stability of GOX@HRP against protease, trypsin was employed as a model digesting enzyme. As shown in Supplementary Fig. 17, free GOX/HRP pairs lost nearly 80% of their activity after trypsin digestion for 2 h at 37 °C. In contrast, GOX@HRP was highly resistant to trypsin digestion, retaining $> 94\%$ of its activity under the same conditions, indicating that the PEG hydrogel-based multi-compartmental particle can effectively protect encapsulated enzymes from biological degradation. Moreover, GOX@HRP is recyclable, as shown in Supplementary Fig. 18. More than 85% of initial activity of GOX@HRP can be retained after five repeated experiments. Overall, the GOX@HRP hydrogel particles are expected to be very useful for mimicking cellular systems.

**pH microenvironment modification in PEG-co-AA hydrogel particles**. As with living cells, there are two main characteristics required to process multistep tandem chemical reactions in one pot[3, 14, 15]. First, close compartmentalization and positional assembly in a micro-sized cell allowed the effective transport of reagents to trigger tandem reactions. Second, different microenvironments (known as organelles) could be generated to spatially confine incompatible opposing reagents in distinguished domains. For example, the pH of the cytosol and the nucleus is $\sim$ 7.2, whereas the lysosome has a pH of 4.7[33].

Previously, the GOX@HRP system was studied to demonstrate the advantage of close compartmentalization and positional assembly in a multi-compartmental system. However, the challenges of performing incompatible reactions remained due to the homogeneous compartments in a system. For example, using GOX and nanozyme as peroxidase mimics for coupled tandem reactions required distinguished compartments with

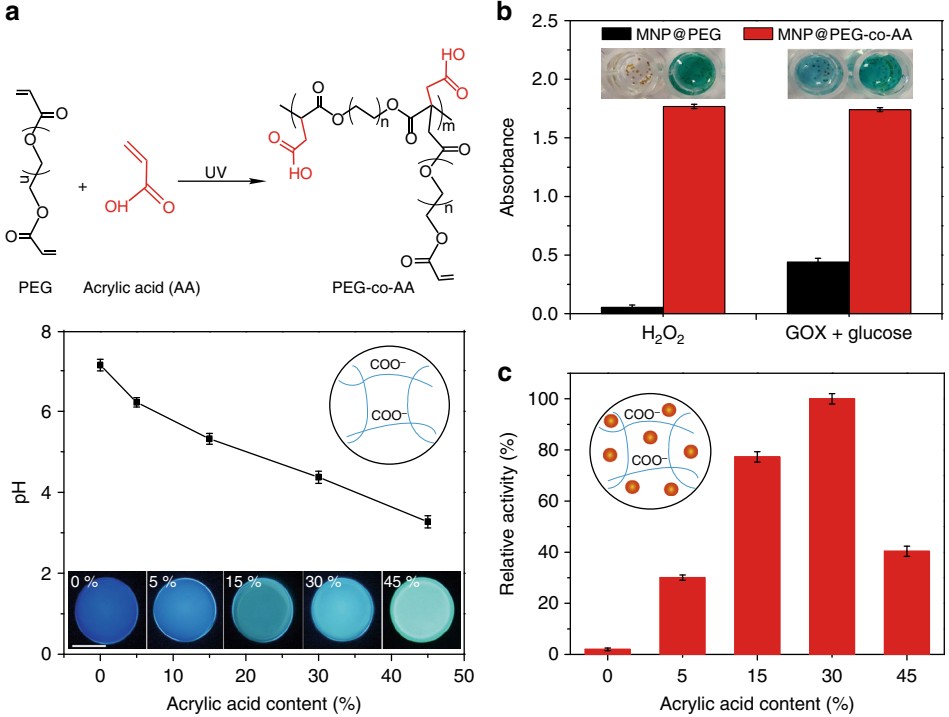

**Fig. 4** AA-mediated pH microenvironments of PEG hydrogels. **a** Schematic representation of the copolymerization of AA with PEG and the pH of the microenvironments of poly(PEG-co-AA) hydrogel particles with different AA contents (0, 5, 15, 30, and 45%). The insets are their corresponding fluorescent photographs. **b** Absorbance at 652 nm and visual colors of oxTMB catalyzed by MNP@PEG (black bars) and MNP@PEG-co-AA (red bars) in PBS (pH 7.0) using $H_2O_2$ as a substrate or coupled with the GOX-catalyzed glucose oxidation reaction. **c** Relative activity changes of MNPs encapsulated in different PEG-co-AA with AA contents ranging from 0 to 45%. Assay conditions: MNP@PEG-co-AA was incubated with 1.25 mM $H_2O_2$ and 43.75 µM TMB in PBS (pH 7.0), and the absorbance was monitored at 652 nm

different pH values. The GOX reaction was processed in a neutral medium (pH ~ 7), while nanozymes should react with $H_2O_2$ in an acidic environment (pH ~ 4). Indeed, to demonstrate the advantage of generating compartments with different pH levels, as commonly witnessed in the cell, it is desirable to manipulate microenvironments in the hydrogel particle to construct a heterogeneous multi-compartmental system with distinct microenvironments for selectively confining incompatible catalysts, allowing an incompatible tandem reaction to be performed in one pot.

To prepare the hydrogel particles with an acidic microenvironment, AA was co-polymerized with PEG[34–36]. To characterize the acidic microenvironments generated, a series of poly(PEG-co-AA) hydrogel particles with AA contents ranging from 0 to 45% were prepared. The LysoSensor conjugated with dextran was encapsulated in a hydrogel particle as a pH indicator to test the pH changes of the interior microenvironment. The existence of conjugated dextran effectively prevented the leakage of the LysoSensor from the hydrogel particle. The LysoSensor is a ratiometric pH probe that produced blue fluorescence in a neutral environment but changes to yellow fluorescence in an acidic environment. As shown in Fig. 4a, with the AA contents increased from 0 to 45%, the poly(PEG-co-AA) hydrogel particles showed obvious color changes shifting from blue to yellow, reflecting their pH changes. By measuring the emission spectra (Supplementary Fig. 19), the pH values of different poly(PEG-co-AA) hydrogel particles were determined, and they ranged from 7.2 to 3.3. Accordingly, poly(PEG-co-AA) hydrogel particles were fabricated to assemble with other hydrogel components through microfluidics to form heterogeneous multi-compartmental hydrogel particles with well-defined pH microenvironment controls.

**MNPs in PEG-co-AA hydrogel particles**. The peroxidase-like activity of the MNPs encapsulated in a poly(PEG-co-AA) hydrogel particle (denoted as MNP@PEG-co-AA) in neutral medium (PBS, 10 mM, pH 7.0, as used below) was investigated to evaluate the feasibility of performing an incompatible catalytic reaction within a poly(PEG-co-AA) hydrogel particle. Carbon-coated $Fe_3O_4$ MNPs with excellent dispersion and stability (Supplementary Fig. 20) were employed as peroxidase mimics[37, 38], and their peroxidase-like activity was determined using the typical TMB colorimetric assay (Supplementary Fig. 21a). Unlike natural peroxidases (e.g., HRP), the peroxidase-like activity of MNP was highly pH dependent, displaying the highest activity at pH 4–5 (Supplementary Fig. 21b). Thus, MNP was used to indicate the effects of the interior environmental pH of a poly(PEG-co-AA) hydrogel particle on the activity of its trapped catalyst.

As shown in Fig. 4b, in the presence of $H_2O_2$, MNP@PEG-co-AA effectively catalyzed the oxidization of TMB in neutral medium, generating oxidized TMB (oxTMB) to show a color change (transparent to blue). In contrast, under the same conditions (neutral medium and in the presence of $H_2O_2$), no clear reaction was observed in MNP@PEG because of the absence of microenvironment controls of pH (Supplementary Fig. 22). The acidic microenvironment generated by AA conjugated to PEG hydrogel particles (MNP@PEG-co-AA) significantly enhanced the catalytic efficiency of MNP (Fig. 4c). The highest activity of MNP@PEG-co-AA was obtained when using MNP@PEG-co-AA with 30% AA. Interestingly, when the AA content reached 45%, the MNP catalysis activity in MNP@PEG-co-AA decreased because the decomposition of MNP occurred at a pH below 4.0[39]. Accordingly, 30% AA was used in the subsequent experiments.

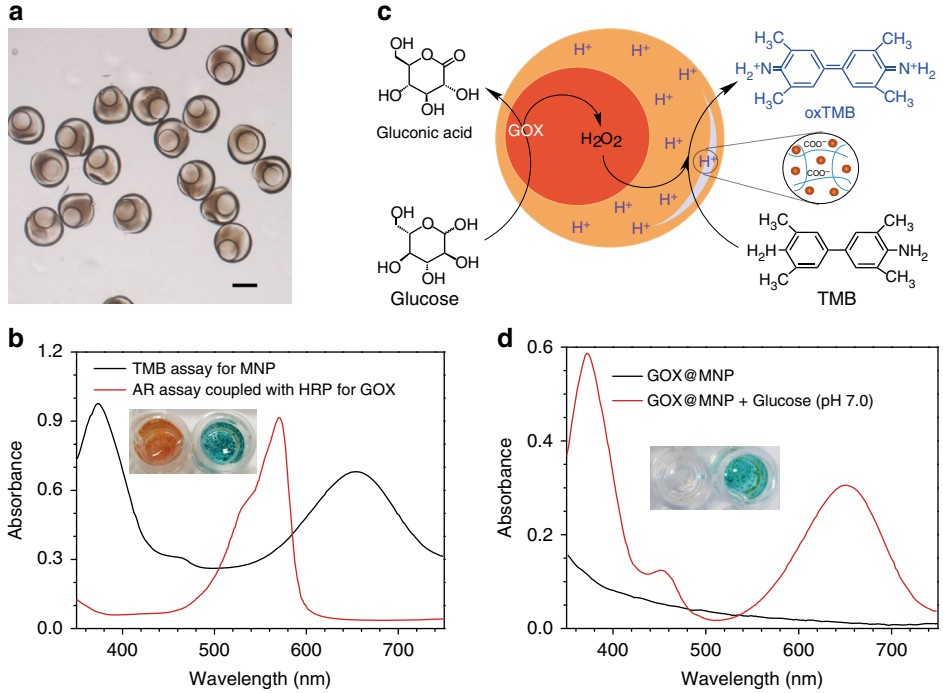

**Fig. 5** GOX@MNP functions as an integrated tandem catalytic system. **a** Microscopic image of GOX@MNP. Scale bar: 200 μm. **b** Absorption spectra of oxTMB (*black line*) and resorufin (*red line*) produced from the GOX@MNP-catalyzed TMB-$H_2O_2$ assay and glucose-triggered Amplex Red assay in the presence of GOX@MNP and HRP, respectively; both were performed in PBS (pH 7.0, 10 mM). *Inset*: corresponding visual colors of these two samples. **c** Schematic illustration of the glucose-triggered TMB oxidation reaction in the presence of GOX@MNP. **d** Absorption spectra and visual color changes of oxTMB obtained from the TMB oxidation reaction catalyzed by GOX@MNP in the absence (*black line*) and presence (*red line*) of glucose in PBS (pH 7.0, 10 mM)

On the basis of the above results, we further exploited the possibility of coupling GOX with MNP@PEG-co-AA to catalyze a glucose-triggered tandem reaction in one pot in natural medium. GOX was denatured at pH 4, losing its catalytic activity[40]. Therefore, GOX and MNP were incompatible, and they were often utilized separately to catalyze single-isolated reactions rather than together in a coupled tandem reaction (Supplementary Fig. 23)[37, 41]. With the advantages in microenvironment controls in MNP@PEG-co-AA, it was found that the TMB solution containing GOX and MNP@PEG-co-AA showed a deep blue color with a strong absorbance of oxTMB after adding glucose (Fig. 4b). In a TMB solution containing GOX and MNP@PEG (without an acid compartment), a weak oxTMB absorbance was observed, which was the result of the self-activated tandem reaction between GOX and MNP[42].

**GOX@MNP system**. The above results demonstrated that the two catalytic reactions involving MNP@PEG-co-AA and GOX could proceed in one pot in neutral medium, mimicking cellular systems. Heterogeneous multi-compartmental hydrogel particles with distinct pH microenvironments were fabricated using PEG hydrogel as the inner compartment and poly (PEG-co-AA) hydrogel as the outer compartment through microfluidics. In addition, to achieve the high-catalytic activity, GOX and MNP were separately confined in the inner and outer compartments of these hydrogel particles, forming an incompatible tandem catalytic system (GOX@MNP) (Supplementary Fig. 24). As shown in Fig. 5a, the morphologies of the as-prepared GOX@MNP particles were similar to those of the GOX@HRP particles.

The activities of GOX and MNP within GOX@MNP were tested in a neutral medium (Fig. 5b). The results not only verified the successful encapsulation of GOX and MNP but also indicated that GOX@MNP exhibited dual enzyme activities in a neutral

medium. These unique features prompted us to explore GOX@MNP as a tandem catalytic system to drive a glucose-triggered incompatible reaction in one pot in a neutral medium. As shown in Fig. 5c, when glucose was added, it was initially oxidized by GOX that was localized in the inner compartment, producing $H_2O_2$, which then moved into the adjacent outer compartment, initiating the subsequent TMB oxidization step mediated by MNPs. Figure 5d shows that a typical oxTMB absorption spectrum and color response can be obtained using GOX@MNP in the presence of glucose, whereas in the absence of glucose, the TMB solution is colorless and shows no absorbance. As the glucose concentration is increased, the absorbance of oxTMB gradually increases, and the TMB solution changes from colorless to deep blue (Supplementary Fig. 25). Therefore, with the assistance of GOX@MNP, glucose can be used to trigger the reaction involved in the TMB colorimetric assay in one pot in a neutral medium.

Subsequently, we investigated the effects of the sub-compartment locations on the overall activity of the GOX and MNP-based tandem catalytic system. To this end, under identical conditions, we fabricated multi-compartmental hydrogel particles containing MNPs in the inner PEG-co-AA compartments and GOX in the outer PEG compartments; these particles are denoted as MNP@GOX (Supplementary Fig. 26). The glucose-triggered TMB colorimetric assay was used to examine the overall activities of GOX@MNP and MNP@GOX. As shown in Supplementary Fig. 27, the overall activity of MNP@GOX was ~ 4-fold lower than that of GOX@MNP, indicating that the sub-compartment locations in hydrogel particles significantly influence the overall activity of these tandem catalytic systems. This finding could be attributed to the loss of intermediate $H_2O_2$ during mass transfer from GOX to MNP, which is greater within MNP@GOX than within GOX@MNP. Because the diffusion

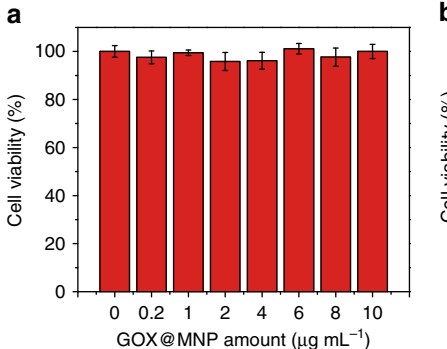

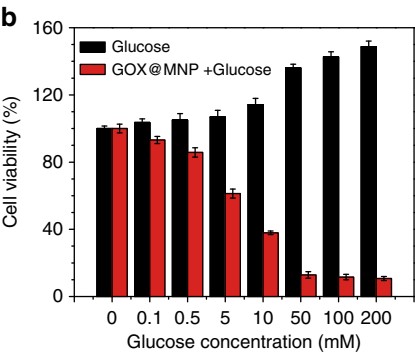

**Fig. 6** Glucose-powered anticancer therapy. **a** Cell viability of HeLa cells after incubation with different concentrations of GOX@MNP for 24 h. **b** Cell viability of HeLa cells after incubation with glucose alone (*black bars*) and with 5 μg ml$^{-1}$ GOX@MNP and different concentrations of glucose (*red bars*) for 24 h

direction of reagents in these multi-compartmental hydrogel particles lacks specificity, once $H_2O_2$ is generated from the GOX-catalyzed oxidization of glucose, it permeates throughout the hydrogel matrix, not just into the intended sub-compartment, resulting in a gradual loss of $H_2O_2$. In contrast, GOX and MNP remain entrapped in the multi-compartmental hydrogel particles because of their larger sizes.

**Glucose-powered therapeutics**. The fluorescence assay of ter-ephthalic acid (TA) demonstrated that the glucose-triggered oxidation of TMB in the presence of GOX@MNP results from the generation of hydroxyl radicals ($^\bullet$OH) through the decomposition of $H_2O_2$, a byproduct of glucose oxidization (Supplementary Fig. 28)[43]. $^\bullet$OH is one of the most reactive oxygen species and can cause oxidative damage to DNA, proteins, and lipids in cells, eventually killing them[44, 45]. Therefore, $^\bullet$OH is highly toxic to cells and is often used to initiate free radical peroxidation in photodynamic and anti-infection therapy[46, 47]. Inspired by this fact, we believe that GOX@MNP can be used to establish a glucose-powered therapy for killing cancer cells. In particular, compared with the direct addition of $H_2O_2$, using glucose as a substrate to produce $^\bullet$OH for therapy is more biofriendly and results in no cytotoxic side effects. Therefore, the biocompatibility of GOX@MNP was first evaluated using a standard 3-(4,5-dimethylthiazol-2-yl)-2,5-diphenyltetrazolium bromide (MTT) assay against HeLa cells. To avoid the influence of glucose, the HeLa cells treated with GOX@MNP were incubated in a glucose-free culture medium (Supplementary Fig. 29). After incubating the cells for 24 h, the viabilities of the HeLa cells treated with GOX@MNP were measured. As shown in Fig. 6a, GOX@MNP showed hardly any adverse effects in the HeLa cells (nearly 100% cell viability), even at concentrations up to 10 μg ml$^{-1}$, indicating the biocompatibility of GOX@MNP. In contrast, in the presence of glucose, GOX@MNP exhibited significant cytotoxicity against the HeLa cells (Fig. 6b). The viabilities of the HeLa cells treated with GOX@MNP decreased as the glucose concentration increased. In addition, > 80% of the HeLa cells treated with GOX@MNP were killed, when 10 mM glucose was added to the culture medium. The half-maximal inhibitory concentration of glucose is estimated as 5.66 mM. Because glucose is essential for cell growth, enhanced cell viabilities were observed among the untreated HeLa cells incubated under the same conditions. Using LIVE/DEAD probes (Supplementary Fig. 30), it was confirmed that almost all of the HeLa cells treated with GOX@MNP in the presence of glucose were dead within 24 h, whereas the control cells remained alive. This result suggests that with the assistance of GOX@MNP, glucose can be used as an anticancer agent for drug-free therapeutics. Moreover, the use of

glucose-powered therapy based on a tandem reaction within GOX@MNP would be a mild, low-cost strategy compared with commonly used chemotherapies. On the basis of this study, the glucose-powered therapy has the potential to emerge as a new modality for cancer therapy. However, the ability of this therapy to selectively kill cancer cells, while maintaining the viability of normal cells requires further improvement. Because these particles are large (diameter ~ 100 μm), as a therapeutic drug, they need to be implanted after surgery instead of being directly injected.

## Discussion

In summary, cell-inspired hydrogel particles with well-defined compartments encapsulating different enzymes in optimized microenvironments were fabricated using microfluidics, enabling the confinement of different contents in distinct domains for processing tandem reactions with high efficiency. The spatial segregation and positional assembly of GOX and HRP in these multi-compartmental hydrogel particles were demonstrated to endow GOX@HRP with ~ 23 times more overall activity than its counterpart, the free GOX/HRP pair. Moreover, when AA was copolymerized with PEG hydrogel, heterogeneous multi-compartmental hydrogel particles with different pH microenvironments were produced. By spatially confining GOX and MNP in defined compartments, the glucose-triggered incompatible multistep tandem reaction could be conducted simultaneously within the particle to produce system-level enzymatic synthetic products, similar to natural cells. Furthermore, based on the high cytotoxicity of hydroxyl radicals, a glucose-powered therapy for eliminating target cancer cells was developed, which may emerge as an advanced cancer therapeutic that does not require the use of toxic drugs. However, the challenges of targeting and killing cancer cells remain. Moreover, because the particles are large, it is necessary to implant them using a surgical method instead of the direct injection method. Nevertheless, in the future, this system could be extended to the desirable one-pot incompatible chemical reactions, resulting in an intelligent, soft system for bio production similar to that of natural cells and drug free long-term treatment. Therefore, this technology is likely to be broadly beneficial to researchers in the fields of biochemical process engineering, bio-fabrication and disease therapeutics.

## Methods

**Fabrication of multi-compartmental hydrogel particle**. The multi-compartmental hydrogel particles were fabricated through a two-step sequential gelation process combined with droplet coalescence in a microfluidic device. The microfluidic device consisted of three T-type polymethylmethacrylate (PMMA) chips connected by a polytetrafluoroethylene tube, as shown in Supplementary

Fig. 1a. Typically, the fabrication procedure began with the formation of gel particles as the inner compartments originating from the gelation of droplets generated from the T 1# chip. Subsequently, the as-formed gel particles flowed into the T 3# chip and further fused with other droplets generated from T 2# (outer compartment), forming hydrogel particles containing two different compartments after treatment with ultraviolet (UV) radiation. Silicon oil without surfactants was used as the continuous phase. A UV filter (OmniCure S1500) was used to provide the desired excitation for polymerization. The flow rates of the dispersed and continuous phases were 2 and 20 µl min$^{-1}$, respectively. Notably, this fabrication method is universal. Multi-compartmental hydrogel particles confining different contents in distinct components can be easily obtained by choosing different pre-gel solutions.

For the homogenous multi-compartmental hydrogel particles, pre-gel solutions I and II contained 20% (v/v) of PEGDA (Mn = 575), 2% (wt%) of the photoinitiator 2-hydroxy-4'-(2-hydroxyethoxy)-2-methylpropiophenone, and one of the following components: water for pure PEG@PEG, fluorescent materials for Albumin-FITC@Dextran-RhB, and enzymes for GOX@HRP. The heterogeneous multi-compartmental hydrogel particles (GOX@MNP) were obtained using two different pre-gel solutions: pre-gel solution I contained 20% (v/v) PEGDA (Mn = 575), 2% (wt%) photoinitiator and 1.5 mg ml$^{-1}$ GOX, whereas pre-gel solution II contained 45% (v/v) PEGDA (Mn = 700), 30% (v/v) AA, 5% (wt%) photoinitiator and 1.2 mg ml$^{-1}$ MNPs. By exchanging the positions of pre-gel solutions I and II, MNP@GOX can be obtained. The total protein contents in these multi-compartmental hydrogel particles were determined using a colorimetric BCA protein assay. The encapsulated GOX amounts were used to calculate the concentrations of GOX@HRP and GOX@MNP in aqueous solutions.

**Fabrication of single-compartmental hydrogel particles**. Single-compartmental hydrogel particles were fabricated using one T-type chip (T1#, Supplementary Fig. 3a). The pre-gel solution contained 20% (v/v) of PEGDA (Mn = 575), 2% (wt%) of photoinitiator and different one of the following: GOX for GOX@PEG, HRP for HRP@PEG, and a mixture of GOX and HRP for GOX&HRP@PEG. The continuous phase was fluorocarbon oil (HFE 7500, 3 M Novec Engineered Fluids) containing 2% (w/w) perfluorinated polyether(PFPE)-PEG block copolymer surfactant. The flow rates were V$_{aqueous}$ = 2 µl min$^{-1}$ and V$_{oil}$ = 20 µl min$^{-1}$. A UV filter (OmniCure S1500) was used to provide the desired excitation for polymerization.

**Overall activities of GOX@HRP and GOX@MNP**. The Amplex Red and TMB colorimetric reactions were used to characterize the overall activities of GOX@HRP and GOX@MNP, respectively. Glucose was used to initiate these two tandem reactions in PBS (10 mM, pH 7.0). The total reaction volumes of these two tandem reactions were 200 µl. After the reactions were complete, the absorption spectra were recorded by a microplate reader. For the detailed procedure for the Amplex Red and TMB colorimetric reactions, please see Supplementary Methods (H$_2$O$_2$-Amplex Red colorimetric reaction for HRP activity and TMB colorimetric assay for MNP activity).

**Kinetic behavior**. The kinetic behavior of GOX@HRP was studied by monitoring the absorbance in 3-min intervals while varying the glucose concentration. The Michaelis–Menten constant was calculated using Lineweaver-Burk plots of the double reciprocal of the Michaelis–Menten equation—$1/\nu = K_m/V_m \, (1/[S] + 1/K_m)$, where $\nu$ is the initial velocity, $V_m$ represents the maximal reaction velocity, [S] corresponds to the substrate concentration, and $K_m$ is the Michaelis constant. Each sample was analyzed three times. All of the error bars represent the standard deviations from three repeated experiments.

**Cell viability and staining**. The standard MTT assay was used to study cell viability. HeLa cells were employed as the model cells and were incubated in Dulbecco's modified eagle medium (DMEM) without glucose. Typically, 100 µl of a HeLa cell solution was pipetted into each well of a 96-well plate to provide a cell density of 5000 cells per well. After incubating the cells for 24 h at 37 °C in a 5% CO$_2$ atmosphere, different amounts of GOX@MNP were added to each well. The plates were incubated for another 24 h, and the cell medium was removed. Subsequently, 10 µl of MTT (12 mM) was added to each well and allowed to react for 4 h at 37 °C. The supernatant in each well was aspirated, and 150 µl of dimethyl sulfoxide was added to each well to dissolve the MTT formazan crystals. Subsequently, the absorbance was determined at 540 nm using a microplate reader (Tecan, Infinite 200 PRO, Switzerland). Cell viability was calculated as the percentage of viable cells after treatment with GOX@MNP compared with the percentage of viable cells among untreated cells. To investigate the glucose-powered killing of cancer cells, the same experimental steps and conditions described above were used, except that different concentrations of glucose were added to the wells containing GOX@MNP.

For cell staining, fluorescein diacetate (FDA) and propidium iodide (PI) were used to stain viable cells and dead cells, respectively. The FDA stock solution was prepared by dissolving 5 mg of FDA in 1 ml of acetone (the stock solution was stored at −20 °C), and the PI stock solution was prepared by dissolving 2 mg of PI in 1 ml of PBS (the stock solution was stored at 4 °C). Briefly, HeLa cells in different wells were first incubated at 37 °C in DMEM without glucose in an atmosphere of 5% CO$_2$ for 24 h. Next, freshly prepared FDA and PI solutions were added and incubated for 5 min at room temperature in the dark. Subsequently, the staining solution was removed, and the cells were washed with PBS. Finally, the as-prepared samples were analyzed using fluorescence microscopy.

**Data availability**. Data supporting the findings of this study are available from the corresponding author on reasonable request.

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

## Acknowledgements

We gratefully acknowledge the funding provided by the NMRC Industry Alignment Fund Category 1 (R-397-000-230-511), NRF BDTA (R-397-000-221-592), MOE Tier-1 (R-397-000-213-112; R-397-000-248-112), A-Star PSF (R-279-000-448-305), and MOE Tier-2 (R-397-000-271-112).

## Author contributions

H.T. and C.-H.C. designed the research. H.T. performed the experiments and data analysis, and contributed to writing the manuscript. S.G. and R.L. contributed to designing the microfluidic device. N.-D.D. contributed to the cell experiments and confocal scanning laser microscopic imaging. L.J. contributed to the simulation of reagent diffusion. C.-H.C. supervised the research and wrote the paper. All authors discussed the results and commented on the manuscript.

## Additional information

**Competing interests:** The authors declare no competing financial interests.

