## [Peer Review File · Nature Communications]

Reviewers' Comments:

Reviewer #1 (Remarks to the Author)

Heterogeneous multi-compartmental hydrogel particles as synthetic cells for incompatible tandem reactions

Hongliang Tan et al

This paper describes a smart multi-compartmentalized system consisting of hydrogelled domains comprising components of two-stage cascade reactions (GOx/HRP; GOx/magnetite), and the influence of the nested confinement on enzyme rates, pH-mediated reactivity and cell toxicity. The work is well performed and clearly presented, although there are sections of the manuscript that appear over-long and pedestrian, and which describe results that are better placed in the SI (controls/Fig 3 etc). In each example presented, the analysis is at a superficial level paying more attention on proof-of-principles than a detailed investigation. However, the concepts described (microfluidic-generated nested droplets, enzyme reactions in multi-compartmentalized (positional) systems (polymersomes for example), proximity effects, peroxide inhibition, artificial peroxidase activity, pH responsive hydrogels) are well known in the mainstream literature, and although the work is of high quality it is not particularly novel or ingenious. In terms of impact, I think at least one of the three aspects of the work (enzyme cascade/pH triggering of enzyme/nanoparticle reactions/cell toxicity) should have been undertaken in much more depth to demonstrate the significance of the work. Overall, I think this manuscript is better placed in a more specialist journal such as ACIE or Chem. Sci.

Specific comments:

1. Kinetic data presented in the Table 1 shows that V_{max} and K_{cat} values obtained for enzymes encapsulated within the gelled compartments are significantly higher than that for free coupled enzyme cascade. This is unusual and needs to be confirmed by more experiments, as it is often observed that the free-system enzyme activities are greater as they are not diffusion limited compared with those in the gelled compartments.
2. The Authors suggest that the products (gluconic acid + H_2O_2) generated from GOx reaction can easily diffuse to the outer gelled compartment containing HRP. However, this mass transfer is not directional. As a result, these products will diffuse in the neighbouring compartments but also will diffuse out of the whole compartment system resulting in considerable dilution of the substrate for HRP reaction. This in turn will affect the rates of reaction, and should be commented on.
3. Recyclability of gelled compartments - It would be interesting to know if the putative high performance is maintained after repeated usage of compartments.
4. The experiment performed to measure the pH of bulk hydrogels (page 12, line 2 from top) is likely to be inaccurate as a polymer layer deposited on to the probe will interfere with the pH values. Instead, encapsulation of pH-indicating dyes would give better control over microenvironmental changes in the pH values.
5. Figure 2 e showing effect of Trypsin seems highly predictable and is better in the SI
6. The Authors claim that OH radicals generated from GOx-MNP system inhibit HeLa cell line. However this should be further supported by measurements of the concentrations of radicals produced during the process. What level of doses are required to curtail the cell growth?
7. Temperature dependent studies (Figure 2d). Along with thermal stabilities of the enzymes, the

authors should also comment on the effect of temperatures on the gels themselves. What are the gel to sol transition temperatures for PEG-diacrylate hydrogels? If it is below 60 degrees C then how does this correlate with observed relative activities? Thermal stability of the gels should be supported by DSC data.

Reviewer #2 (Remarks to the Author)

In this report a microfluidic strategy is used to create a microgel in microgel system, in which different catalysts are positioned in the different compartments. The authors use the well-known GOx-HRP couple and extend this to combining GOX with catalytic nanoparticles. Finally the cascade gels are used for killing cells by the creation of radicals.

Although there is certainly a technological interest in this paper, the concepts presented are not novel. A range of multicompartiment systems is known now, and they have been used for tandem catalysis of also incompatible enzymes. Controlling the microenvironment around a catalyst to direct its activity has also been presented before. Finally, GOx-loaded nanoreactors have been shown even to work in cells for the production of radicals (see Tingholm et al Small 2016).

Technical comments:

Although release is not high, it is also not negligible (10% after 24 h). As release occurs, so does diffusion through the microgels, which leads to mixing of the enzymes in the different compartments. The authors should analyse this, for example by labelling the enzymes with a fluorescent probe.

The proximity effect only plays a real role in submicron compartments. The particles used here are multi-micrometer in size and as such diffusion distances are still large. Furthermore, it is confusing that the authors on the one hand claim that proximity plays a role, and on the other say that in the microgel case there is no build-up of hydrogen peroxide, because this has to diffuse to the next microgel. Hydrogen peroxide, due its small size and neutral character, will be hardly affected by a hydrogel environment, and this cannot be an explanation of a minimized inhibitory effect. To investigate more clearly the tandem reaction, the authors should look at the catalytic activity of the two enzymes separately, and not determine the overall k_{cat} and V_{max} of the entire system to investigate in which step the increase in activity takes place.

How do the authors explain the results in fig 2c and S8? In the latter figure, there is no difference in activity between the free and encapsulated enzymes. In Fig 2 the encapsulation leads to a twofold increase.

To test the microenvironment's pH the authors should incorporate a pH sensitive dye.

For a therapeutic application, the particles are much too big, and is therefore not realistic.

Fig S9 is very difficult to read.

Reviewer #3 (Remarks to the Author)

The manuscript by Chen et al. describes the preparation of compartmentalized hydrogels and their application as cell-like mimics, in which an enzyme/enzyme mimic is placed in each of the compartments and is used to carry out multi-step incompatible tandem reactions. The hydrogels are prepared using microfluids. In the first step, a drop of a specified size is generated that contains water, a UV photoinitiator, a desired enzyme(s), and PEG-diacrylate as a monomer capable of forming a highly crosslinked hydrogel. Exposure of this mixture to UV radiation creates a rigid hydrogel with a particular enzyme anchored within. In a second similar step, a larger drop is formed around the original droplet, this time containing a different enzyme. Further addition of UV radiation is thus able to generate a "gel within a gel" structure, as repeatedly confirmed via microscopy. Following the preparation of these hydrogels, the authors use them in three different enzymatic applications. In the first application, glucose oxidase (GOX) is placed in the smaller compartment while horseradish peroxidase (HRP) is placed in the larger encapsulating compartment (using the notation GOX@HRP to indicate that GOX is in a smaller compartment-

surrounded by a larger compartment containing HRP). This prepared hydrogel can catalyze a two-step process, in which glucose is first oxidized by GOX, forming gluconic acid and hydrogen peroxide. The hydrogen peroxide generated is then catalyzed by HRP in the second step, in a process in which Amplex Red is oxidized to form resorufin, a compound whose red color can be easily detected and quantified. This process, which occurs in the compartmentalized hydrogel, is compared to three control setups- 1) a tandem reaction performed by free GOX and free HRP, 2) a tandem reaction in which GOX and HRP are both confined to separate hydrogels, and 3) a tandem reaction in which GOX and HRP are both confined to the same, non-compartmentalized hydrogel. The compartmentalized hydrogel exhibits vastly greater activity in the two step process over any of three controls. When free GOX and HRP are used, the large amount of hydrogen peroxide generated in the first step can inhibit the activity of HRP. The authors speculate that by compartmentalizing the two enzymes in isolated, yet spatially proximal domains, the products of the first reaction can easily diffuse to the second enzyme, ensuring high overall activity.

In the second application, HRP is replaced in the second step with an iron containing magnetic nanoparticle (MNP), which has been shown to have peroxidase-like activity. However, unlike HRP, the MNP is highly pH dependent and is most active at a pH of around 4. Therefore, the outer compartment containing the MNP was formed using a mixture of PEG-diacrylate and acrylic acid to generate an acidic environment. Incorporation of acrylic acid is critical, since encapsulating MNP in an environment that solely consists of the PEG hydrogel will greatly diminish its catalytic performance. A similar two step tandem reaction as described above was performed, in which the HRP was replaced with the MNP in a step that oxidizes TMB to form oxTMB. Placing GOX in the smaller compartment appears to be critical, since switching the location of the two enzymes/catalysts lowers the overall activity. The authors speculate that this phenomenon is caused by the random diffusion exhibited by hydrogen peroxide in this process (in the initial process hydrogen peroxide is generated in the smaller compartment and thus can only diffuse outward into the larger compartment).

In the third and final application, the GOX@MNP hydrogel is evaluated for its potentially cytotoxic behavior. The authors hypothesize that the generation of hydroxyl radicals in the tandem process can be utilized for their highly cytotoxic behavior. The authors show that without glucose present the GOX@MNP hydrogel displays no cytotoxic behavior, but the addition of glucose causes a dramatic increase in cytotoxicity. They suggest that such a system could be attractive since it is triggered by the initial addition of harmless glucose (as opposed to a more harmful species like hydrogen peroxide) although they admit that such a system lacks cell specificity.

Compartmentalization is currently an extremely active and exciting topic in various fields of chemistry-the ability to mimic the cell's ability to perform multiple competing reactions simultaneously has drawn considerable interest from both a material science and organic chemistry perspective. This paper demonstrates that a relatively simple and easy to synthesize hydrogel can replicate this highly advantageous feature of the cell. Although the author's focus is on the GOX→HRP tandem reaction (and derivatives thereof), in theory such a system could be applied to any number of tandem reactions in the cell that employ reagents that inhibit one another. Therefore, because of the broad impact of this work, I recommend that this manuscript be accepted into Nature Communications. The paper has high impact from a biological, chemistry and engineering standpoint. Furthermore, I was impressed by the plethora of control reactions run by the authors, proving the highly advantageous features endowed by a compartmentalized setup. Although the paper as a whole is of high quality, a few small clarifications would be helpful.

- It would be prudent for the authors to give some context into the GOX/HRP incompatible tandem sequence-this is an extremely popular system due to the incompatibility of the two steps and as such has been used to evaluate many different compartmentalized systems. Furthermore, the authors would be wise to mention the 2007 work from the van Hest group, in which GOX and HRP are two steps of a three step tandem process.

- The motivation for switching from HRP to the MNP, about halfway through the manuscript, could be described in a manner that is more relevant to the work. Here, the authors only mention "superior properties relative to natural enzymes" without specifying what those problems actually. Addressing a certain issue with the first part of the work (in which GOX and HRP are used) and explaining how switching from HRP to MNP would solve such an issue, would provide for a cleaner transition. As currently written, the decision to switch from HRP to MNP seems somewhat random. One possibility is that the authors could discuss the desire to generate compartments with differing pH's, as commonly witnessed in the cell. Since the authors do indeed need MNP-containing compartment to have a lower pH, this would make the motivation for this section clearer.

- Typo. In Page 14, In the context "As shown in Fig. S16, the overall activity of GOX@MNP was approximately 4-fold lower than that of GOX@MNP", the first GOX@MNP should be MNP@GOX.

- What will happen if changing the compartmental particle size ratio or loading ratio of the two enzymes (in the paper, it is roughly 1:1), given that the author claims that higher local conc. of H₂O₂ from GOX&HRP@PEG can inhibit the HRP activity.

Replies to Reviewers' Comments

Reviewer #1:

This paper describes a smart multi-compartmentalized system consisting of hydrogelled domains comprising components of two-stage cascade reactions (GOx/HRP; GOx/magnetite), and the influence of the nested confinement on enzyme rates, pH-mediated reactivity and cell toxicity. The work is well performed and clearly presented, although there are sections of the manuscript that appear over-long and pedestrian, and which describe results that are better placed in the SI (controls/Fig 3 etc). In each example presented, the analysis is at a superficial level paying more attention on proof-of-principles than a detailed investigation. However, the concepts described (microfluidic-generated nested droplets, enzyme reactions in multi-compartmentalized (positional) systems (polymersomes for example), proximity effects, peroxide inhibition, artificial peroxidase activity, pH responsive hydrogels) are well known in the mainstream literature, and although the work is of high quality it is not particularly novel or ingenious. In terms of impact, I think at least one of the three aspects of the work (enzyme cascade/pH triggering of enzyme/nanoparticle reactions/cell toxicity) should have been undertaken in much more depth to demonstrate the significance of the work. Overall, I think this manuscript is better placed in a more specialist journal such as ACIE or Chem. Sci.

Answer: We appreciate the referee's comment. The multi-compartmental platforms were important to effectively perform a wide range of essential multistep tandem reactions, generating desirable products. Recently, various chemical systems, such as liposomes, polymersomes and polyelectrolyte capsules, for the compartmentalization of biochemical reactions have been developed to carry out different cascade reactions in one pot (*Angew. Chem. Int. Ed.* **2007**, *46*, 5605; *Angew. Chem. Int. Ed.* **2009**, *48*, 4359; *Langmuir* **2012**, *28*, 13798; *Angew. Chem. Int. Ed.* **2014**, *53*, 146). However, the contradiction between effective intermediate reagent transportation and micro-environmental controls remained. For example, liposomes and polymersomes were developed to immobilize enzymes triggering a cascade process, but the challenges of controlling individual, different compartmental microenvironments (such as pH) remained (*Small* **2013**, *9*, 3573; *J. Mater. Chem. B* **2014**, *2*, 2733; *Nat. Commun.* **2014**, *5*, 5305). The Pickering emulsions and functional polymeric micelles were manipulated to form different compartments to optimize the incompatible tandem reactions, but these synthetic materials required organic chemical processes for their preparation, causing challenges in biocompatibility (*J. Am. Chem. Soc.* **2015**, *137*, 1362; *J. Am. Chem. Soc.* **2015**, *137*, 12984). Moreover, without well-defined close packing, the product generation efficiency was limited by the slow transportation between the compartments in Pickering emulsions and polymeric micelles.

There are two main reasons resulting in the high efficiency of generating bio-products within living cells. First, close compartmentalization and positional assembly in a micro-sized cell allowed effective transportation of reagents for reactions. Second, different microenvironments (such as different pH levels) were generated to enable incompatible opposing reagents to be spatially confined in distinguished domains. Consequently, numerous multistep reactions could occur simultaneously in a living cell with unsurpassed efficiency and specificity (*Curr. Opin. Plant Biol.* **2004**, *7*, 254; *Nat. Rev. Mol. Cell Biol.* **2010**, *11*, 50; *Adv. Mater.* **2016**, *28*, 1109). For example, the Calvin cycle for carbon fixation, which involves multiple enzymes and substrate molecules, can occur in multiple successive cellular compartments with different microenvironments; this process takes place in mitochondria but has interactions with many cytosolic pathways. The confinement of multiple enzyme reactions in the incompatible compartments to mimic living

cells still remains to be exploited. Although functional hydrogels have been widely demonstrated to manipulate different microenvironments, these studies have focused on the applications of enzyme immobilization, reagent release and drug delivery (*Adv. Mater.* **2002**, *14*, 743; *Soft Matter* **2009**, *5*, 707; *Angew. Chem. Int. Ed.* **2013**, *52*, 13538; *Nature Rev. Mater.* **2016**, *1*, 16071).

In this study, a new cell-mimicking system using different spatial-confining enzymes (GOX and HRP) in a multi-compartmental hydrogel particle was approached using physical encapsulation through microfluidics. The combination of distinguished hydrogel compartments and enzyme immobilization is synergetic. The dense package of hydrogels in micro-scaled allowed effective transportation of reactants, while the distinguished compartments offered desirable micro-environment controls to ensure product generation with high efficiency. For example, the positional assembly of GOX and HRP in this micro-scaled multi-compartmental particle (GOX@HRP) was demonstrated to enhance reaction activity to generate resorufin (an approximate 23-fold enhancement) compared with that of the free GOX/HRP coupled cascade system. Upon the copolymerization of acrylic acid, the incompatible reactions catalyzed by GOX and MNP can be carried out in this multi-compartmental particle in one pot with promising efficiency, which is difficult to access using the previously reported multi-compartmental systems. Moreover, the physical encapsulation of hydrogel building blocks through microfluidics, without relying on specific chemical procedures, allows the presented technology to effectively generate a wide range of products that are required in cascade tandem reactions as a novel universal platform.

To offer a detailed investigation of the cell-mimicking multi-compartmental hydrogel particles, a series of additional experiments on the characterization of the enzyme cascade reactions were included in the revised manuscript. The catalytic reaction of GOX/HRP (the first aspect of the work) was specifically investigated in-depth. To identify the location of different hydrogel compartments in a particle, the positions of GOX and HRP were analyzed using a confocal scanning microscope after labelling GOX and HRP with RBITC and FITC, respectively (Fig. 2b). The reagent transportation/diffusion profiles of GOX and HRP were indicated quantitatively by the absorbance of RBITC and FITC (Fig. 2c). By individually measuring the kinetic behaviors of GOX and HRP in GOX@HRP (Table S1), it was found that the K_m and turnover number K_{cat} of GOX and HRP in GOX@HRP were similar to those of free suspended enzymes (control samples), reflecting the use of the free diffusion of intermediates in the multi-compartmental particle to enable the processing of cascade reactions in GOX@HRP. In our experiments, a significant enhancement of catalytic activity of the GOX and HRP coupled reaction systems was observed using multi-compartmental particles (GOX@HRP).

To investigate the reagent transportation between compartments, the time-dependent fluorescent changes of Amplex Red were investigated in the presence of GOX@HRP and the free GOX/HRP systems. A 3-min lag phase before a fluorescence increase was observed from the free GOX/HRP system compared with that of GOX@HRP (Fig. S12), indicating the effective transport of the intermediates among the enzymes due to the close packing of GOX and HRP in a multi-compartmental hydrogel particle. Similar phenomena to those observed with Amplex Red were also observed in the colorimetric reactions using 3,3',5,5'-tetramethylbenzidine (TMB) and 2,2'-azino-bis(3-ethylbenzothiazoline-6-sulphonic acid) (ABTS) as substrates (Fig. S13). It was found that the locations of the hydrogel compartments with immobilized enzymes were important to trigger effective cascade reactions.

In addition, the inhibition of H_2O_2 on HRP activity in the reaction was observed. H_2O_2 inhibition was minimized in a multi-compartmental particle because of the spatial segregation of GOX and HRP, leading to a high catalytic activity of GOX@HRP. To investigate the mechanism in detail, various GOX@HRP and GOX&HRP@PEG systems with a molar ratio of GOX to HRP from 1:1 to 10:1 were used to catalyze glucose-initiated tandem reactions (Fig. 3d and S14). It was found that with the same diffusion distance, a high molar ratio of GOX to HRP generated high local concentrations of H_2O_2 , causing low activity of the GOX and HRP coupled reaction system. Therefore, in summary, the positional assembly and spatial segregation of GOX and HRP in a micro-scale multi-compartmental system worked synergistically to perform cascade tandem reactions. H_2O_2 produced from the GOX reaction in the inner compartment was effectively transferred to the outer compartment to trigger the HRP reaction, while the HRP activity was not suppressed due to the direct interaction with a high concentration of H_2O_2 . An additional discussion was added on P.9-10 in the revised manuscript.

Specific comments:

1. Kinetic data presented in the Table 1 shows that V_{max} and K_{cat} values obtained for enzymes encapsulated within the gelled compartments are significantly higher than that for free coupled enzyme cascade. This is unusual and needs to be confirmed by more experiments, as it is often observed that the free-system enzyme activities are greater as they are not diffusion limited compared with those in the gelled compartments.

Answer: As stated in the referee's comment, based on the adjustable properties of a polymeric network of hydrogels, the reagent diffusion profile could be regulated to allow small molecules to pass freely through the polymer matrix, while large macromolecules (e.g., enzymes) remain trapped inside (*Adv. Mater.* **2002**, *14*, 743; *Anal. Chem.* **2005**, *77*, 6828). In this work, the intermediate (H_2O_2) was neutral in charge. In this case, the size of H_2O_2 was ~ 0.4 nm, which was much smaller than the mesh size of the hydrogel particle (*Eur. Polym. J.* **2015**, *72*, 386). Accordingly, H_2O_2 can freely cross the polymeric network of hydrogels, showing a similar diffusion profile of that of H_2O_2 in a free system.

Our experimental investigations on the catalytic activity (Fig. S4 in the revised manuscript) indicated that the diffusion of substrates (glucose and H_2O_2) in a free enzyme (GOX or HRP) solution was similar to that in enzyme (GOX or HRP)-immobilized hydrogels. We also characterized the kinetic behaviors of the separation of GOX and HRP in GOX@HRP (Table S1). It was found that the individual V_{max} and K_{cat} values of GOX and HRP in GOX@HRP were similar to those of free GOX and HRP, supporting that the diffusion of the intermediate (H_2O_2) in GOX@HRP was not limited by the hydrogel matrix.

Based on the close packing of individual compartments in our multi-compartmental system, the effective transport of H_2O_2 was observed, resulting in a higher activity of GOX@HRP than that of the free GOX/HRP system. To characterize the mechanism, we recorded the time-dependent fluorescence change of Amplex Red in the presence of GOX@HRP and in the presence of the free GOX/HRP system. The results in Fig. S12 showed a 3-min lag phase before a fluorescence increase in the free GOX/HRP system compared with the fluorescence increase in GOX@HRP. This result indicated that the fluorescence increase of the free GOX/HRP system was significantly slower than that of GOX@HRP, which was due to the time required for H_2O_2 to transfer from GOX to HRP for the reactions. Similar patterns to those observed with Amplex red were

observed with GOX@HRP involved in reactions using TMB and ABTS as substrates (Fig. S13), demonstrating the potential of the presented system as a universal platform. The activity of GOX@HRP was further compared to the GOX@PEG/HRP@PEG system without the proximity effect (Fig. 3c). It was found that the activity of the GOX@PEG/HRP@PEG system was lower than that of GOX@HRP. A series of additional experimental results were added in the revised manuscript, P. 9-10.

2. The Authors suggest that the products (gluconic acid + H₂O₂) generated from GOx reaction can easily diffuse to the outer gelled compartment containing HRP. However, this mass transfer is not directional. As a result, these products will diffuse in the neighbouring compartments but also will diffuse out of the whole compartment system resulting in considerable dilution of the substrate for HRP reaction. This in turn will affect the rates of reaction, and should be commented on.

Answer: Many thanks for the comment. Indeed, H₂O₂ was free to diffuse within a GOX@HRP particle (not directional) due to its small size and neutral charge. It was not easy to accurately measure the amounts of H₂O₂ that diffused out from the whole compartmental system to the bulk solution. Therefore, to discuss the effects of the H₂O₂ amount on the GOX and HRP coupled reactions in a multi-compartmental system, a series of glucose conversion experiments catalyzed by various GOX@HRP amounts with molar ratios of GOX to HRP from 1:1 to 10:1 were conducted, as the local H₂O₂ amount was determined by the GOX concentrations (Fig. 3d and Fig. S14a). It was found that as the GOX amounts increased in GOX@HRP, the oxidation reactions of Amplex Red were stopped rapidly (Fig. S14a). Moreover, a decreased turnover number of GOX@HRP was observed upon the increase of the GOX concentration (Fig. 3d). These results confirmed the inhibition effect of a high local concentration of H₂O₂ on HRP activity (*Enzyme Microb. Tech.* **1997**, *21*, 302; *Chem. Biol.* **2002**, *9*, 555; *RSC Adv.* **2014**, *4*, 37244) and suggested that the dilution of the local H₂O₂ concentration at an HRP site due to free diffusion was important to the enhancement of GOX@HRP activity. Additional experimental results were added in the revised manuscript, P. 9-10.

3. Recyclability of gelled compartments - It would be interesting to know if the putative high performance is maintained after repeated usage of compartments.

Answer: The recyclability/stability of GOX@HRP was investigated accordingly. It was found that more than 85 % of the initial activity of GOX@HRP was retained after five repeated experiments (Fig. S18). Moreover, compared with the free GOX/HRP system, long-term enzyme stability in GOX@HRP in a water solution was observed (Fig. S15). After incubating at 25 °C for 7 days, the GOX@HRP retained ~85% of the initial activity, while the free GOX/HRP system lost ~40% of the initial activity within a 1-day incubation. Additional results were added in the revised manuscript, P. 10-11.

4. The experiment performed to measure the pH of bulk hydrogels (page 12, line 2 from top) is likely to be inaccurate as a polymer layer deposited on to the probe will interfere with the pH values. Instead, encapsulation of pH-indicating dyes would give better control over microenvironmental changes in the pH values.

Answer: Based on referee's comment, to measure the pH values in each compartment, encapsulation of pH-indicating dyes in the compartments was performed. The dye employed was LysoSensor (Thermo Fisher Scientific, L22460) with dextran conjugation to test the pH changes. This dye showed a blue fluorescence in

neutral environments but showed yellow fluorescence in acidic environments. The fluorescent photos and emission spectra of the LysoSensor encapsulated in the poly(PEG-co-AA) hydrogel with different AA amounts in the revised manuscript are shown in Fig. 4a and S19.

5. Figure 2e showing effect of Trypsin seems highly predictable and is better in the SI

Answer: Fig. 2 was revised accordingly. The original Fig. 2e was moved to Fig. S17 in the SI.

6. The Authors claim that OH radicals generated from GOx-MNP system inhibit HeLa cell line. However this should be further supported by measurements of the concentrations of radicals produced during the process. What level of doses are required to curtail the cell growth?

Answer: We appreciate the constructive comment. In the third aspect of this work, a drug-free cell inhibition method through $\cdot\text{OH}$ generation by glucose degradation is presented in the GOX@MNP system. Instead of direct measurement of the $\cdot\text{OH}$ concentration to suppress HeLa cell growth, the inhibitory concentration of glucose was measured. The two reasons for using the glucose concentration to evaluate the cell inhibitory efficiency of the GOX@MNP system were, first, the $\cdot\text{OH}$ concentration in our multi-compartmental system could not be precisely measured due to the free diffusion of H_2O_2 produced from the GOX reaction (using glucose as the substrate). Second, HeLa cell growth was suppressed by both H_2O_2 and OH radicals due to their toxicity to the cells (*FEMS Microbiol. Rev.* **2013**, 37, 955; *ACS Nano* **2014**, 8, 6202). With the measurements of cell viabilities under various glucose concentrations in the GOX@MNP system, the half maximal inhibitory concentration (IC50) of glucose to suppress HeLa cell growth was estimated as ~ 5.66 mM. An additional discussion was added in the revised manuscript, P.17.

7. Temperature dependent studies (Figure 2d). Along with thermal stabilities of the enzymes, the authors should also comment on the effect of temperatures on the gels themselves. What are the gel to sol transition temperatures for PEG-diacrylate hydrogels? If it is below 60 degrees C then how does this correlate with observed relative activities? Thermal stability of the gels should be supported by DSC data.

Answer: Thank you for the comment. In this work, the poly(PEG) hydrogel polymerized through UV-induced polymerization of poly(ethylene glycol) diacrylate (PEGDA) was used for enzyme encapsulation. During the polymerization, the C=C bond of PEGDA was converted to a C-C bond in the poly(PEG) hydrogel, forming a stable chemical bond. The thermal stability of the poly(PEG) hydrogel was confirmed by DSC analysis in **Figure R1** below. The results showed that there were two endothermic reactions during the heating process of the poly(PEG) hydrogel. The first endothermic peak was attributed to the loss of water, while the second peak corresponded to the sol transition temperature of the poly(PEG) hydrogel at 127 °C, which was higher than the thermal stability temperature in our experiments (60 °C). Moreover, the experimental results in long-term enzyme stability in the GOX@HRP system at 25 °C also showed that there were no effects of the poly(PEG) hydrogel on the enzyme activity (Fig. S15). Additional experimental results were included in the revised manuscript, P. 10.

Figure R1. Differential scanning calorimetry (DSC) curve of the poly (PEG) hydrogel.

Reviewer #2:

In this report a microfluidic strategy is used to create a microgel in microgel system, in which different catalysts are positioned in the different compartments. The authors use the well-known GOx-HRP couple and extend this to combining GOx with catalytic nanoparticles. Finally, the cascade gels are used for killing cells by the creation of radicals. Although there is certainly a technological interest in this paper, the concepts presented are not novel. A range of multicompartement systems is known now, and they have been used for tandem catalysis of also incompatible enzymes. Controlling the microenvironment around a catalyst to direct its activity has also been presented before. Finally, GOx-loaded nanoreactors have been shown even to work in cells for the production of radicals (see Thingholm et al Small 2016).

Answer: As stated in the referee's comment, a range of multi-compartmental systems (liposomes, polymersomes, water-in-oil emulsions, and polyelectrolyte capsules) were developed to perform various tandem reactions, including tandem catalysis of incompatible reagents. For example, liposomes were investigated to construct vesicle-based artificial cells as chemical microreactors for multi-step enzymatic reactions (*Nat. Commun.* **2014**, *5*, 5305). The functional polymers were developed as transition metal catalyst supports for incompatible catalytic transformations in one pot (*J. Am. Chem. Soc.* **2015**, *137*, 12984). Although several multi-compartmental systems were developed previously to process the cascade reactions, these systems were limited regarding the effective intermediate reagent transportation, micro-environmental controls and unpredictable living cell product generation.

In this study, a new cell-mimic multi-compartmental system by spatial confining different enzyme (GOx and HRP) was approached through microfluidics. There are two important features for a multi compartmental system to perform cascade tandem catalysis of incompatible reagents. Firstly, compartments should be closely assembled with well-defined position in a micro-sized chamber to allow effective transportation of intermediate reagents for the reactions. Secondly, different microenvironments (such as different pH) should be generated to enable incompatible opposing reagents to be spatially confined in distinguished domains. The combination of distinguished hydrogel compartments and enzyme immobilization through microfluidics is synergetic. The dense package of hydrogels in micro-scaled allowed effective transportation of reactants, while the distinguished compartments offered desirable micro-environment controls to ensure product generation

with high efficiency.

The nano-carriers were developed to deliver target reagents to interact with cells for the product generation. For example, the HRP or GOX-loaded nanoreactors were demonstrated to interact with living cells for the production of radicals (*Angew. Chem. Int. Ed.* 2010, 49, 7213; *Adv. Funct. Mater.* 2014, 24, 4625; *Small* 2016, 12, 1806). This technology can also be applied to deliver DNA/reagent to stimulate the organelles for product generation as applications in synthetic biology (*Angew. Chem. Int. Ed.* 2009, 48, 329; *Chem. Eur. J.* 2011, 17, 4552; *Angew. Chem. Int. Ed.* 2016, 55, 11377). However, the generation of products from these nanoreactors relied on living cell environment and the permeability of carriers themselves, which reached limitations in stable product yielding, unpredictable side cellular products and intensive culturing process (*Chem. Sci.* 2012, 3, 335).

It is worth to note that our presented therapy strategy is different from therapeutic GOX-loaded nanoreactor reported by Thingholm et al. The presented therapy strategy was based on glucose-triggered incompatible multistep tandem reaction, which was performed in a multi-compartmentalized structure with accurate controls as synthetic cells. In addition, by using presented multi-compartmental system, the reaction product used for killing cell was hydroxyl radical rather than H₂O₂ presented before by Thingholm et al. Compared with H₂O₂ produced by living cells stimulated by GOX-loaded nanoreactor, our multi compartmental hydrogel based systems generating hydroxyl radical in cell-free environment with precise concentration controls showed better efficiency to suppress/kill target cells (*FEMS Microbiol. Rev.* 2013, 37, 955; *ACS Nano* 2014, 8, 6202).

Technical comments:

1. Although release is not high, it is also not negligible (10% after 24 h). As release occurs, so does diffusion through the microgels, which leads to mixing of the enzymes in the different compartments. The authors should analyse this, for example by labelling the enzymes with a fluorescent probe.

Answer: Thank you for the comment. To characterize the spatial distribution/diffusion of GOX and HRP in GOX@HRP multi-compartmental hydrogel particles, each type of enzyme was labelled with a specific fluorescence probe. The GOX (core) and HRP (shell) were labelled with RBITC ($\lambda_{\text{ex}} = 555 \text{ nm}$, $\lambda_{\text{em}} = 625 \text{ nm}$) and FITC ($\lambda_{\text{ex}} = 490 \text{ nm}$, $\lambda_{\text{em}} = 525 \text{ nm}$), respectively. The confocal scanning laser microscopy images (Fig. 2b) indicated the distinct domains of the compartments in GOX@HRP in which the GOX-loaded compartment was positioned in the inside of the HRP-loaded compartment. The individual release of GOX and HRP in the GOX@HRP system was quantitatively measured using the UV absorbance of RBITC and FITC (Fig. 2c). It was observed that the individual release of GOX and HRP was less than 5 % within 24 hours, suggesting that GOX and HRP were well confined in the individual compartments of this hydrogel particle. An additional discussion was included in the revised manuscript, P.6. The original Fig. S10 has been replaced by the new Fig. 2c.

2. The proximity effect only plays a real role in submicron compartments. The particles used here are multi-micrometer in size and as such diffusion distances are still large. Furthermore, it is confusing that the authors on the one hand claim that proximity plays a role, and on the other say that in the microgel case there is no

build-up of hydrogen peroxide, because this has to diffuse to the next microgel. Hydrogen peroxide, due its small size and neutral character, will be hardly affected by a hydrogel environment, and this cannot be an explanation of a minimized inhibitory effect.

Answer: We appreciate the comment. The proximity effect plays an important role in submicron compartmental systems. However, although the diffusion distance of the presented multi-micrometer system was longer than that of submicron compartmental systems, the experimental results showed that the proximity effect was still important to perform effective tandem reactions in a multi-micrometer system. A 23-fold enhancement in GOX@HRP compared with the free GOX/HRP system was observed.

Indeed, H₂O₂ did not manipulate the hydrogel microenvironment, and the changes in the local concentration of H₂O₂ showed a significant influence on the HRP activity. In several previous studies, a high local concentration of H₂O₂ suppressed the HRP activity (*Enzyme Microb. Tech.* **1997**, *21*, 302; *Chem. Biol.* **2002**, *9*, 555; *RSC Adv.* **2014**, *4*, 37244). Therefore, to achieve a high reaction activity, there are two requirements for the GOX and HRP coupled cascade reaction: 1). The distance between GOX and HRP should be close enough to allow the effective transfer of H₂O₂, thus triggering the cascade reactions. 2). A certain distance between GOX and HRP should be maintained to ensure an HRP with high activity for the oxidation of substrates (*Nat Nano.* **2009**, *4*, 249). Accordingly, the micro-scale compartmental system was an ideal platform to perform GOX and HRP coupled cascade reactions to ensure H₂O₂ effective transport for triggering the cascade reaction, while minimizing the inhibition effect of H₂O₂ on HRP.

To characterize the proximity effect and the H₂O₂ inhibition effect, a series of additional experiments were conducted. First, we recorded the time-dependent fluorescence change of Amplex Red in the presence of GOX@HRP and the free GOX/HRP system. A 3-min lag phase for the reaction (indicated by an increase in fluorescence) in the free GOX/HRP system was observed compared to the reaction occurring in GOX@HRP (Fig. S12). As the GOX in GOX@HRP has a comparable affinity and reaction rate to those of glucose compared with free GOX (Table S1), the slower conversion of chromogenic substrates catalyzed by the free GOX/HRP system was attributed to the time required for the hydrogen peroxide transfer from GOX to HRP for the reaction. Moreover, the free GOX/HRP system involved in the reactions using TMB and ABTS as substrates also exhibited a slower increase in the absorbance of the reduction products than that of GOX@HRP (Fig. S13). This result was consistent with previous reports (*Chem. Commun.* **2014**, *50*, 12465; *Nat. Commun.* **2016**, *7*, 10619).

Second, various GOX&HRP@PEG and GOX@HRP systems with a molar ratio of GOX to HRP from 1:1 to 10:1 to catalyze glucose-initiated tandem reactions (Fig. 3d and S14) were applied to show the reaction activities under different local concentrations of H₂O₂. It was found that with the same diffusion distance, a high molar ratio of GOX to HRP generated high local concentrations of H₂O₂ that caused low activity of the GOX and HRP coupled reaction system. Our results suggested that packaging GOX and HRP in a multi-compartmental micro-scaled system was important to ensure the transport of the intermediate H₂O₂ to trigger cascade reactions, as well as to minimize H₂O₂ inhibition of HRP due to direct interactions. Additional results and a discussion can be found in the revised draft, P. 9-10.

3. To investigate more clearly the tandem reaction, the authors should look at the catalytic activity of the two enzymes separately, and not determine the overall K_{cat} and V_{max} of the entire system to investigate in which step the increase in activity takes place.

Answer: The catalytic activity of the two enzymes was characterized using a set of additional control experiments, as requested. We investigated the reaction kinetics of separate GOX and HRP in GOX@HRP particles to compare with the reaction kinetics in the free GOX system and the free HRP system. The results showed that the K_{cat} and V_{max} values of separate GOX and HRP in GOX@HRP particles were similar to the K_{cat} and V_{max} values of free GOX and HRP. This result indicated that the high catalytic activity in GOX@HRP resulted from the spatial segregation and positional assembly of GOX and HRP in a multi-compartmental particle. The additional new experimental results are included in Table S1 in the revised manuscript.

4. How do the authors explain the results in Fig 2c and S8? In the latter figure, there is no difference in activity between the free and encapsulated enzymes. In Fig 2 the encapsulation leads to a two-fold increase.

Answer: The enzyme distributions in the two reaction systems (the free system and encapsulated system) were different. To compare the enzyme activity of the encapsulated system with that of the free system in Fig. S8a (new Fig. S4a in the revised manuscript), the total enzyme contents in these two systems were set to be the same within the same reaction volume. For the free system, the enzymes were freely dissolved in a solvent, forming a homogeneous mixture. However, in the case of the encapsulated system, the enzymes were immobilized in the hydrogel particles with a high local concentration, and the particles were dispersed in a solution.

In the case of single-type particle systems (GOX@PEG or HRP@PEG particle, new Fig. S4a), although the local enzyme concentration in a particle was high, the particles were highly dispersed, leading to a long distance between particles (due to the same amount of enzyme in a fixed reaction volume). Therefore, the reaction activity of an individually encapsulated system was similar to that of free system.

In the case of mixing two types of particles to form the GOX@PEG/HRP@PEG system (Fig. 2c, new Fig. 3c in the revised manuscript), a fixed amount of GOX@PEG particles and HRP@PEG particles were mixed in a fixed reaction volume to determine the amount of GOX and HRP, respectively. Accordingly, the total particle number (including GOX@PEG particles and HRP@PEG particles) was 2-fold higher than the total particle number in a single-type particle system in the same reaction volume. This situation leads to closer contact of the GOX@PEG particles and HRP@PEG particles, resulting in the effective transport of H_2O_2 for the reactions. Therefore, in Fig. 2c (new Fig. 3c in the revised manuscript), the activity with a ~2-fold enhancement in the GOX@PEG/HRP@PEG system compared with the activity in the free GOX/HRP system was observed.

5. To test the microenvironment's pH the authors should incorporate a pH sensitive dye.

Answer: To test the pH in different microenvironments, the pH-sensitive dye LysoSensor (Thermo Fisher Scientific, L22460) conjugated with dextran was encapsulated in the hydrogel compartment, as requested,

which showed blue fluorescence in a neutral environment, while showed yellow fluorescence in an acidic environment. The poly(PEG-co-AA) hydrogel particles showed obvious color changes from blue to yellow as the acrylic acid contents increased from 0 to 45 % (Fig. 4a), reflecting the pH changes of these particles. By measuring their emission spectra (Fig. S19), the pH values of these poly(PEG-co-AA) hydrogel particles were determined to range from 7.2 to 3.3. Additional results were included in the revised manuscript, Fig. 4a and S19.

6. For a therapeutic application, the particles are much too big, and is therefore not realistic.

Answer: The size of the fabricated multi-compartmental hydrogel particles was ~300 μm . Unlike nano particles/sub-micro particles, these microsized multi-compartmental hydrogel particles were not suitable for direct injection into the bloodstream for treatments. Instead of direct particle injection into the bloodstream, with the advantage of drug-free therapeutics, these particles could be implanted in the injured area/subcutaneous fat through an injection/operation for long-term therapeutics. Recently, a series of micro-sized hydrogel particles were developed and demonstrated promising clinical applications with encouraging therapeutic outcomes (*Adv. Funct. Mater.* **2012**, *22*, 3793; *Adv. Mater.* **2016**, *28*, 3669; *Biomaterials* **2017**, *113*, 170).

7. Fig S9 is very difficult to read.

Answer: Fig. S9 in the original manuscript was re-plotted with a high resolution. Fig. S9 in the original draft is now Fig. S11 in the revised manuscript.

Reviewer #3:

- It would be prudent for the authors to give some context into the GOX/HRP incompatible tandem sequence- this is an extremely popular system due to the incompatibility of the two steps and as such has been used to evaluate many different compartmentalized systems. Furthermore, the authors would be wise to mention the 2007 work from the van Hest group, in which GOX and HRP are two steps of a three step tandem process.

Answer: We appreciate the referee's comments. As the GOX/HRP sequence is a popular system to characterize compartmentalized systems (*Org. Biomol. Chem.* **2008**, *6*, 4315; *Chem. Eur. J.* **2009**, *15*, 1107; *Biomacromolecules* **2010**, *11*, 1480; *J. Mater. Chem. B* **2014**, *2*, 2733; *Nat. Commun.* **2014**, *5*, 5305), a discussion to elaborate GOX/HRP reactions was added on P.5 in our revised draft. The work by the van Hest group to introduce two-step GOX and HRP for a three-step tandem process (*Angew. Chem. Int. Ed.* **2007**, *46*, 7378) was also discussed and was cited in the revised draft, Ref 7.

- The motivation for switching from HRP to the MNP, about halfway through the manuscript, could be described in a manner that is more relevant to the work. Here, the authors only mention "superior properties relative to natural enzymes" without specifying what those problems actually. Addressing a certain issue with the first part of the work (in which GOX and HRP are used) and explaining how switching from HRP to MNP would solve such an issue, would provide for a cleaner transition. As currently written, the decision to switch from HRP to MNP seems somewhat random. One possibility is that the authors could discuss the desire to generate compartments with differing pH's, as commonly witnessed in the cell. Since the authors do indeed need MNP-containing compartment to have a lower pH, this would make the motivation for this section clearer.

Answer: Many thanks for the constructive comment. There are two advantages of the presented cell-mimicking multi-compartmental hydrogel particles through microfluidics: First, close compartmentalization and positional assembly in a micro-sized cell allowed the effective transport of reagents for reactions. Second, different microenvironments (such as different pH levels) could be generated to spatially confine incompatible opposing reagents in distinguished domains. In the first part of the work, the GOX@HRP system was studied to show the advantage of close compartmentalization and positional assembly of GOX and HRP in a micro-sized cell for the effective transport of intermediates. However, GOX and HRP reactions were performed in the compartments with the same microenvironments (pH = 7.2), so this cascade reaction could not be used to demonstrate the advantage of generating compartments with different pH levels, as commonly observed in the cell. To address this issue, it is desirable for us to switch from HRP to MNP. Therefore, in the second work, we investigated the incompatible reactions of GOX and MNP, which require a system containing multiple compartments with different pH microenvironments to be performed in one pot. The GOX reaction was performed in an inner compartment (pH = 7.2) generating H₂O₂. Then, H₂O₂ was transferred to the outer compartment to react with HRP (pH = 4.4) and to oxidize TMB to oxTMB to give a blue color. The motivation for switching HRP to MNP was discussed in the revised manuscript, P. 11-12.

- Typo. In Page 14, In the context “As shown in Fig. S16, the overall activity of GOX@MNP was approximately 4-fold lower than that of GOX@MNP”, the first GOX@MNP should be MNP@GOX.

Answer: The typo has been corrected in the revised manuscript.

- What will happen if changing the compartmental particle size ratio or loading ratio of the two enzymes (in the paper, it is roughly 1:1), given that the author claims that higher local conc. of H₂O₂ from GOX&HRP@PEG can inhibit the HRP activity.

Answer: Different loading ratios of the two enzymes, GOX and HRP, in a multi-compartmental system compared with GOX&HRP@PEG with the same enzyme contents, were investigated accordingly to characterize the effect of the local concentration of H₂O₂ on HRP activity. GOX@HRP and GOX&HRP@PEG with different molar ratios of GOX to HRP from 1:1 to 10:1 were fabricated using microfluidics. The results (Fig. S14a) showed that as the molar ratio of GOX to HRP increased from 1:1 to 10:1, the initial reaction rates of GOX&HRP@PEG decreased significantly, suggesting the existence of an inhibition effect of local H₂O₂ on HRP activity in GOX&HRP@PEG. Similarly, the inhibition of GOX@HRP by H₂O₂ produced from glucose oxidation was verified by the suppression of the oxidation of Amplex Red upon the increase of GOX amounts (Fig. S14b). Nevertheless, it was found that the turnover numbers of GOX@HRP were much higher than that of GOX&HRP@PEG under identical conditions (Fig. 3d), indicating that the inhibition of H₂O₂ (on the HRP activity) in GOX@HRP was weaker than that in GOX&HRP@PEG. Additional results were included in the revised manuscript, P. 9-10.

Reviewers' Comments:

Reviewer #1:

Remarks to the Author:

The authors have worked hard to develop the paper, which now includes more in-depth analysis and a clearer presentation of the work in both the main text and SI. The responses were thoughtful and detailed, and significant changes and new work have been included. However, it would have been easier to assess the changes if the authors had highlighted their modifications in the revised documents, rather than just referring to them generally in the response letter.

One outstanding problem is Fig 1 and the caption, both of which seem to be left unfinished (no labelling; no parts in the caption), so it was difficult to discern what the different parts of the figure refer. Similarly, the accompanying text in the manuscript seems to be very inconsequential when it comes to describing why Fig 1 is relevant.

Overall, given a final minor editing by the authors, I am now happy to recommend the work for publication. It is high class work and will be of wide ranging interest.

Reviewer #2:

Remarks to the Author:

The revised paper shows some improvements compared to the original version. However, the level of novelty is still not sufficient for what would be expected of Nature Communications.

Furthermore, the effect of compartmentalization remains elusive. Enzyme activities have not changed, but the GOx@HRP system seems much more efficient. I am not convinced that this is a positional effect due to the large size of the microenvironments. The authors should calculate diffusion profiles and diffusion gradients in their systems to substantiate their points regarding efficiency in the different systems. I can't see why a GOx@HRP system would lead to less inhibition.

Comparison with compartmentalization in the cellular environment is not appropriate, as these are nanometer sized structures.

Therapeutic applications remain unrealistic even after the explanation by the authors.

Technical comments:

Cumulative release mentioned in the text is higher than shown in figure 1 (5 vs 10%). It is not clear what causes this difference, even when these data concern different enzyme hydrogels.

Cross contamination is still not clearly quantified.

There seems to be a discrepancy in the results in fig S11, when one compares the inset in fig S11d with the other data.

Reviewer #3:

None

Replies to Reviewers' Comments

Reviewer #1:

The authors have worked hard to develop the paper, which now includes more in-depth analysis and a clearer presentation of the work in both the main text and SI. The responses were thoughtful and detailed, and significant changes and new work have been included. However, it would have been easier to assess the changes if the authors had highlighted their modifications in the revised documents, rather than just referring to them generally in the response letter.

Answer: Many thanks for the comments. The modifications were highlighted by yellow lines in our new revised manuscript.

One outstanding problem is Fig 1 and the caption, both of which seem to be left unfinished (no labelling; no parts in the caption), so it was difficult to discern what the different parts of the figure refer. Similarly, the accompanying text in the manuscript seems to be very inconsequential when it comes to describing why Fig 1 is relevant.

Answer: We are appreciated by referee's constructive comment and detailed checking. Figure 1 was revised to include labelling of sub-figures. The caption and main text related with Figure 1 were revised with ordered manner accordingly.

Overall, given a final minor editing by the authors, I am now happy to recommend the work for publication. It is high class work and will be of wide ranging interest.

Reviewer #2:

1. The revised paper shows some improvements compared to the original version. However, the level of novelty is still not sufficient for what would be expected of Nature Communications.

Answer: In this study, a new cell-mimetic system using different spatial-confining enzymes (GOX and HRP) in a multi-compartmental hydrogel particle was approached using physical encapsulation of microfluidics. Although several multi compartmental systems, such as liposomes, polymersomes, and polymer capsules were reported before, the contradiction between close confinements with dense packing and micro-environmental controls of different compartments remained (*Small* 2013, 9, 3573; *J. Mater. Chem. B* 2014, 2, 2733; *Nat. Commun.* 2014, 5, 5305; *J. Am. Chem. Soc.* 2015, 137, 1362; *J. Am. Chem. Soc.* 2015, 137, 12984). The combination of distinguished hydrogel compartments and enzyme immobilization in a well-defined micro-sized hydrogel particle via microfluidic assembly is synergetic. The dense package of hydrogels in micro-scaled allowed effective transportation of reactants, while the distinguished compartments offered desirable micro-environment controls to ensure product generation with high efficiency.

To address the novelty of this system, the positional assembly of GOX and HRP in this micro-scaled multi-compartmental particle (GOX@HRP) was approached to enhance reaction activity to generate resorufin (23-fold enhancement) compared with that of the free GOX/HRP coupled cascade system. Upon the copolymerization of acrylic acid, the incompatible reactions catalyzed by GOX and MNP can be carried out in this multi-compartmental particle in one pot with promising efficiency, which is difficult to access using the previously reported systems. Moreover, the physical encapsulation of hydrogel building blocks through microfluidics, without relying on specific chemical procedures, allows the presented hydrogel system to effectively generate a wide range of enzymatic products required cascade tandem reactions, as a novel universal platform.

2. Furthermore, the effect of compartmentalization remains elusive. Enzyme activities have not changed, but the GOx@HRP system seems much more efficient. I am not convinced that this is a positional effect due to the large size of the microenvironments. The authors should calculate diffusion profiles and diffusion gradients in their systems to substantiate their points regarding efficiency in the different systems.

Answer: The efficiency to perform a cascade enzymatic reaction is affected by two factors: 1. enzyme activities of individual step-reactions, and 2. the transportation efficiency of intermediate product to trigger the sequential following-step reactions. In GOX@HRP system, although enzyme activities of individual step reactions have not been changes, high efficiency of intermediate product (H₂O₂) transportation was approached due to closed packing of GOX and HRP in microscale (positional effect).

There are two advantages of close packing in GOX@HRP system. Firstly, the diffusion distance of H₂O₂ from GOX to HRP was decreased (~100 μm), compared with the distance of GOX@PEG + HRP@PEG system (~500 μm). Secondly, the GOX core was fully covered by HRP shell (Fig. 2b), forming large contact area of H₂O₂ to allow effective HRP reaction. Accordingly, once GOX reaction (the first step reaction) was performed by adding glucose, the following HRP reaction (the second step reaction) was triggered rapidly to produce resorufin with high efficiency. It is worth to note that positional effect based on diffusion gradient is not the physical phenomenon only occurred in nanoscale. According to Fick's laws of diffusion, positional

effect can be observed in micro-scale as well.

To clarify the positional effect in presented micro-sized multi compartmental systems, additional simulation results were conducted to calculate diffusion profiles of H_2O_2 , shown in **Response Fig. 1** below. Brownian motion model was used to estimate H_2O_2 diffusion gradient between two compartments (*J. Am. Chem. Soc.* 2012, 134, 5516; *Nat. Chem.* 2016, 8, 299), which triggered the following HRP reaction to produce resorufin. In this simulation, it was found that without dense packing of GOX and HRP (GOX@PEG + HRP@PEG system), the distance between GOX@PEG hydrogel particle (first step enzyme reaction) and HRP@PEG hydrogel particle (second step enzyme reaction) was $\sim 500 \mu\text{m}$. Therefore, when H_2O_2 diffused to HRP@PEG, the concentration decreased to $\sim 20\%$ of the initial concentration of H_2O_2 (assume the GOX reaction occurred in the center of the core), causing limited efficiency to produce resorufin. While in GOX@HRP system, with dense packing of GOX and HRP in a hydrogel particle via microfluidics, the distance between GOX hydrogel component (core part, first step enzyme reaction) and HRP hydrogel component (shell part, second step enzyme reaction) was $\sim 100 \mu\text{m}$. Therefore, the H_2O_2 concentration remained $\sim 90\%$ of the initial concentration of H_2O_2 to effectively trigger following HRP enzyme reaction, generating resorufin. The additional discussion was included in P.9 in the revised article and Fig. S13 in the revised supporting information.

Response Fig. 1: H_2O_2 concentration gradient as a function of distance with following parameters: diffusion coefficient $\sim 1000 \mu\text{m}^2/\text{s}$, K_{cat} (in GOX reaction) $\sim 300 \text{s}^{-1}$, and the integration time $\sim 100 \text{s}$. The model of H_2O_2 diffusion is calculated by using Brownian motion model reported before (*J. Am. Chem. Soc.* 2012, 134, 5516; *Nat. Chem.* 2016, 8, 299).

3. I can't see why a GOX@HRP system would lead to less inhibition.

Answer: In GOX&HRP@PEG system, GOX and HRP were homogenously mixed together in a hydrogel particle. In this system, H_2O_2 generated by GOX reaction (first-step reaction) directly interacted with HRP to suppress HRP reaction activity (second-step reaction), which was resulted from the high local concentration of H_2O_2 (*Enzyme Microb. Tech.* 1997, 21, 302; *Chem. Biol.* 2002, 9, 555; *RSC Adv.* 2014, 4, 37244). In a

GOX@HRP system, H₂O₂ generated in GOX reaction (first step reaction) did not interact with HRP directly, because of distinguished compartmentalization of GOX (core) and HRP (shell) in a well-defined structured hydrogel particle. Therefore, H₂O₂ generated in core part needed to diffuse to shell part and to interact with HRP. Accordingly, the concentration of H₂O₂ decreased, leading to less inhibition. To verify this effect, a series of experiments in glucose conversion by using GOX@HRP systems and GOX&HRP@PEG systems (with molar ratios of GOX to HRP from 1:1 to 10:1) were conducted, as shown in Fig. 3d and Fig. S14.

4. Comparison with compartmentalization in the cellular environment is not appropriate, as these are nanometer sized structures.

Answer: With the advantage of dense packing (like a cell), in a multi-compartmental hydrogel particle, the distance to transport the intermediate from one compartment to another is regulated. There are two main reasons resulting in the high efficiency of generating bio-products within living cells. First, close compartmentalization and positional assembly in a micro-sized cell allowed effective transportation of reagents for reactions. Second, different microenvironments (such as different pH levels) were generated in individual organelles to enable incompatible opposing reagents to be spatially confined in distinguished domains. Consequently, numerous multistep reactions could occur simultaneously in a living cell with unsurpassed efficiency and specificity. In the presented cell mimetic multi-compartmental hydrogel system, both advantages, close compartmentalization (effective intermediate transportation) and micro-environmental controls, were demonstrated to trigger incompatible tandem reactions generating chemical products (resorufin) with high efficiency, as what a living cell could approach.

5. Therapeutic applications remain unrealistic even after the explanation by the authors.

Answer: With the capability of triggering cascade enzymatic reactions by using a GOX@MMP system, a new concept of drug free (glucose powered) therapeutics to kill cancer cells was presented. The added glucose was firstly converted to gluconic acid and H₂O₂ through a GOX-mediated oxidation reaction. After that, the formed H₂O₂ reacted with MNP to produce [•]OH to kill HeLa cells. The efficiency was characterized by using MTT assay and LIVE/DEAD staining, as shown in Fig. 6 and Fig. S30. It was found that the cell viability decreased with increasing glucose concentration. The half-maximal inhibitory concentration of glucose was determined as 5.66 mM. More than 80 % HeLa cells treated with GOX@MNP in the presence of 10 mM glucose were killed within 24 h. This is a new/novel therapeutic method for long term cancer cell elimination without uploading any drugs, which clinical efficiency will be future evaluated.

Technical comments:

1. Cumulative release mentioned in the text is higher than shown in figure 1 (5 vs 10%). It is not clear what causes this difference, even when these data concern different enzyme hydrogels. Cross contamination is still not clearly quantified.

Answer: In our original manuscript, the releasing of GOX and HRP in a GOX@HRP system was determined by using bicinchoninic acid (BCA) assay. The measured releasing rate (9.9%) was the sum of GOX and HRP releasing. In the revised manuscripts, to quantify the releasing of individual GOX and HRP in specific

compartments (GOX@HRP system), GOX and HRP were labelled by different fluorescent dyes, RBITC (emission wavelength 555 nm) and FITC (emission wavelength 490 nm), respectively. The results (Fig. 2c) showed that the GOX release rate was ~4.8 % and HRP release rate was ~4.3 %, which suggest that both GOX and HRP were well confined in the distinguished compartments with limited cross contamination. The total release rate (9.1 %) of GOX and HRP measured by the absorbance of labelled dyes was consistent with the total releasing rate (9.9%) measured by BCA assay.

2. There seems to be a discrepancy in the results in fig S11, when one compares the inset in fig S11d with the other data.

Answer: Many thanks for the comment. A typo in unit was found in Fig. S11d (new Fig. S10d in the revised manuscript). The velocity unit in Y-axis should be 10^{-7} M s^{-1} . The new Fig. S10d was included in the revised supporting information.

Reviewers' Comments:

Reviewer #2:

Remarks to the Author:

The authors have addressed the comments raised by the reviewers in an adequate manner. The additional information on diffusion provides a rationale why the system could work better as multicompartment microgel. One aspect I still disagree with is the biomedical potential of these large particles. The authors should tone down their expectations with regard to the applicability of these materials in this context.

Replies to Reviewer's Comments

Reviewer #2:

The authors have addressed the comments raised by the reviewers in an adequate manner. The additional information on diffusion provides a rationale why the system could work better as multicompartment microgel. One aspect I still disagree with is the biomedical potential of these large particles. The authors should tone down their expectations with regard to the applicability of these materials in this context.

Answer: Many thanks for the comments. The modifications in context to tone down the expectations in potential biomedical applications were highlighted by yellow lines in our new revised manuscript.